

**LA-ICP-MS U-Pb carbonate geochronology: strategies, progress, and**
**application to fracture-fill calcite**
Nick M W Roberts[1], Kerstin Drost[2], Matthew S A Horstwood[1], Daniel J Condon[1], , David
Chew[2], Henrik Drake[3], Antoni E Milodowski[4], Noah M McLean[5], Andrew J Smye[6], Richard J
Walker[7], Richard Haslam[4], Keith Hodson[8], Jonathan Imber[9], Nicolas Beaudoin[10]
[1]Geochronology and Tracers Facility, British Geological Survey, Environmental Science
Centre, Nottingham, NG12 5GG, UK
[2]Department of Geology, Trinity College Dublin, Dublin 2, Ireland
[3]Department of Biology and Environmental Science, Linnaeus University, 39231 Kalmar,
Sweden
[4]British Geological Survey, Environmental Science Centre, Nottingham, NG12 5GG, UK
[5]Department of Geology, University of Kansas, Lawrence, KS 66045, USA
[6]Department of Geosciences, Pennsylvania State University, University Park, PA 16802,
USA
[7]School of Geography, Geology, and the Environment, University of Leicester, Leicester,
LE1 7RH, UK
[8]Department of Earth and Space Sciences, University of Washington, Seattle, WA 98195,
USA
[9]Department of Earth Sciences, Durham University, Science Labs, Durham, UK
[10]Laboratoire des Fluides Complexes et leurs Réservoirs-IPRA, E2SUPPA, Total, CNRS,
Université de Pau et des Pays de l'Adour, UMR5150, Pau, France



**Abstract**
Laser Ablation Inductively Coupled Plasma Mass Spectrometry (LA-ICP-MS) U-Pb
geochronology of carbonate minerals, calcite in particular, is rapidly gaining popularity as
an absolute dating method. The technique has proven useful for dating fracture-fill calcite,
which provides a powerful record of palaeohydrology, and within certain constraints, can be
used to bracket the timing of brittle fracture and fault development. The high spatial
resolution of LA-ICP-MS U-Pb carbonate geochronology is beneficial over traditional
Isotope Dilution methods, particularly for diagenetic and hydrothermal calcite, because
uranium and lead are heterogeneously distributed on the sub-mm scale. At the same time,
this can provide limitations to the method, as locating zones of radiogenic lead can be time-
consuming and 'hit or miss'. Here, we present strategies for dating carbonates with in situ
techniques, through imaging and petrographic techniques to data interpretation; we focus
on examples of fracture-filling calcite, but most of our discussion is relevant to all carbonate
applications. We demonstrate these strategies through a series of case studies. We review
several limitations to the method, including open system behaviour, variable initial lead
compositions, and U-daughter disequilibrium. We also discuss two approaches to data
collection: traditional spot analyses guided by petrographic and elemental imaging, and
image-based dating that utilises LA-ICP-MS elemental and isotopic map data.
**1. Introduction**
Calcite ($CaCO_3$), along with other carbonate minerals (e.g. aragonite, dolomite, magnesite),
forms in a wide variety of geological environments as both a primary and secondary mineral
phase, including diagenetic, biogenic, igneous, metamorphic and hydrothermal
environments. Calcite can incorporate uranium upon its formation, making it a potentially
suitable chronometer for U-Pb and U-Th geochronology. Calcite geochronology therefore
has the potential to provide direct timing constraints to a broad suite of geoscience
applications. Calcite has been dated in the past by chemical dissolution and isotope dilution
(ID) with measurement by either Thermal Ionisation Mass Spectrometry (TIMS) or
Inductively Coupled Plasma Mass Spectrometry (ICP-MS) (e.g. Smith and Farquhar, 1989;
DeWolf and Halliday, 1991; Brannon et al., 1996; Rasbury et al., 1997; Richards et al.,
1998; Woodhead et al., 2006; Pickering et al., 2010), collectively referred to here simply as
Isotope Dilution (ID). More recently, there has been a proliferation in the use of laser



ablation (LA-) ICP-MS applied to calcite geochronology (Li et al., 2014; Coogan et al., 2016;
Roberts & Walker, 2016, Ring & Gerdes, 2016; Methner et al., 2016; Goodfellow et al.,
2017; Burisch et al., 2017, 2018; Drake et al., 2017; Hansman et al., 2017; Hellwig et al.,
2018; Godeau et al., 2017; Beaudoin et al., 2018; Drost et al., 2018; Mangenot et al., 2018;
Nuriel et al., 2017, 2019; Parrish et al., 2018; Walter et al., 2018; Smeraglia et al., 2019;
Holdsworth et al., 2019; MacDonald et al., 2019; Scardia et al., 2019). Presently, we are not
aware of successful secondary ion mass spectrometry (SIMS) U-Pb dating of carbonate
mineralisation, but this presents an alternative microbeam method to LA-ICP-MS.

The first review of the possibilities for carbonate geochronology was published by Jahn &
Cuvellier (1984), and this was substantially updated by Rasbury & Cole (2009). The latter
provided up-to-date discussion on U-Pb isotope systematics in carbonates, particularly
regarding Pb-Pb and U-Pb isochron methods, as well as a review of the applications to
date. At that time, both marine- (e.g. limestone, dolomite) and meteoric-water sourced
carbonates (e.g. speleothems and tufas) had received the most attention, due to their often-
favourable uranium contents, and studies of hydrothermal carbonate were scarce (e.g.
Brannon et al., 1996; Grandia et al., 2000). U-Pb dating of speleothems has been further
reviewed by Woodhead et al. (2006 and 2012), again, focussing on data generated by ID.

Now that microbeam (i.e. LA-ICP-MS and SIMS) U-Pb geochronology is proving to be a
useful method for a range of geoscience applications, it is pertinent to address what can be
achieved with the method, what the current limitations are, and where improvements can be
made in the future. We refer to LA-ICP-MS through the rest of this paper, but acknowledge
that nearly all of the points we cover are equally relevant to SIMS methods. The key benefit
to LA-ICP-MS dating is that its high spatial resolution can be used to relate U-Pb and other
geochemical analyses to imaged textures. This is critical for providing context to the
obtained dates. Carbonate materials are heterogeneous in composition elementally,
isotopically, and texturally. These factors can all lead to scatter in U-Pb data, and will often
hinder the ability to generate high precision (i.e. <1% $2\sigma$) U-Pb dates. In fact, after
propagation of all relevant uncertainties, final U-Pb dates typically exceed 3% precision
($2\sigma$). For this reason, LA-ICP-MS carbonate U-Pb geochronology is particularly suited for
applications in tectonics and crustal fluid-flow, but commonly less suited for applications in
stratigraphy and palaeoclimate.



Here we present a review of LA-ICP-MS U-Pb carbonate geochronology, focusing on its
benefits and application, with particular attention to hydrothermal and diagenetic vein-filling
carbonates; these can constrain the ages of mineral systems, crustal deformation and fluid-
flow, and represent a significant growth area for this method. Using several case studies,
we highlight the utility of image-guided analysis, where various imaging techniques provide
critical context for interpreting U-Pb data. We also provide case studies for an age-mapping
technique that is an alternative to traditional static spot ablation, and can be used in
combination with sample imagery to generate U-Pb age data. Finally, we highlight issues
surrounding initial lead compositions, initial disequilibrium in the U-Pb system and open-
system behaviour.

## 112   2. LA vs ID strategies

Geochronology by ID provides the most accurate assessment of the U-Pb age of a sample
using calibrated isotopic tracer solutions, but it is time-consuming and requires a clean
laboratory facility for sample dissolution and column chemistry. The spatial resolution of ID
is typically much lower than that offered by microbeam techniques, although resolution can
be increased by using a high precision micro-drill for direct sampling. A major limiting factor
is that carbonate materials typically have very low U concentrations (ca. 10 ppb to 10 ppm
U) compared with traditional U-bearing accessory minerals (e.g., often >100 ppm U in
zircon). This means that: 1) comparatively large volumes of material are needed for ID
analyses resulting in an 'averaging' effect and reduction of spread in U/Pb space, and 2)
samples with lower Pb concentrations yield higher blank/sample ratios, hindering the
accuracy and precision of the resulting data.

LA-ICP-MS is a much quicker technique than ID, and therefore less expensive per analysis.
Several samples can be run in a single day, meaning the technique is ideal for screening of
large sample sets to find the most suitable material. The effect of blanks is negated, and
very low (<100 ppb) Pb contents can be analysed. However, LA-ICPMS is generally less
precise analytically compared to ID approaches. Another major limitation is the need to
normalise to a matrix-matched reference material. This means that the uncertainty of the
reference material becomes a limiting uncertainty, and matrix effects between materials of
different composition will generate scatter and/or bias in the U-Pb dates that are difficult to
correct for.




The biggest benefit of LA-ICP-MS comes from the spatial resolution (less than ca. 100 μm)
at which data can be obtained, particularly given the length scales of uranium concentration
heterogeneity in carbonate. We find that for hydrothermal and diagenetic calcite in
particular, uranium is heterogeneously distributed across veins and vein phases, and within
individual crystals (see Figure 1). Uranium concentration heterogeneity typically spans 1 to
3 orders of magnitude, with the length-scale of this variation being commonly much less
than 1 mm. Targeting of high U domains is therefore difficult without a high spatial-
resolution sampling method. Intracrystalline uranium distributions within calcite define
several patterns (see Figure 1): concentrated along cleavage planes (A), growth-zone
controlled (C, D and F), concentrated towards grain rims (areas of B and E), and with
apparent disorder (areas of B and E). Laser ablation has the spatial resolution capable of
targeting such elemental (and isotopic) zonation, making it easier to avoid distinguishable
alteration zones and inclusions at the 10-100 μm scale.



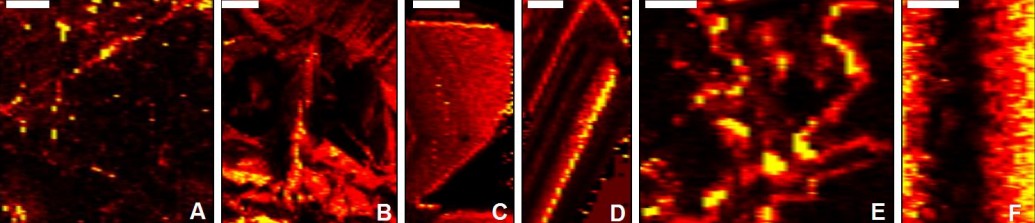

*Figure 1. Maps of uranium in vein-filling calcite from a range of geological*
*settings showing varying styles of distribution, see text for explanation (brighter*
*= higher concentration). Maps were generated using LA-ICP-MS trace element*
*analyses and the Iolite data reduction software. Scale bar is 1 mm.*


For minerals such as zircon, U-Pb ID-TIMS is considered the gold standard of
geochronological techniques (Renne et al., 1998). It offers significantly greater accuracy
than microbeam techniques by virtue of the use of gravimetrically quantified isotopic spikes,
and generally higher detection efficiencies. ID-TIMS does, however, consume greater
amounts of material. With ID methods, ages are calculated by absolute determination of the





number of atoms of each isotope in the sample material. In contrast, microbeam techniques
are relative methods, using ratio normalisation against reference minerals of known
composition (generally determined by ID methods).


For common-lead bearing minerals such as calcite, the extreme range in parent/daughter
ratios encountered (quoted here as $^{238}$U divided by initial lead as $^{204}$Pb; a ratio known as μ),
means that ID does not always lead to an improvement in precision on the regressed age.
This is demonstrated by the schematic model in Figure 2. Sampling for ID provides an
average of elemental and isotopic zonation within the analytical volume, perhaps >1 mm$^3$,
depending on the concentration of U and Pb within the crystal(s). The resulting data should
be precise (depending on the sample/blank ratios), but may potentially have a small spread
in parent/daughter ratios (i.e. $^{238}$U/$^{206}$Pb) due to the averaging effect during sampling. In
contrast, LA sampling has the potential to target and utilise such zonation, better resolving
end-member μ compositions, and resulting in analyses with a greater spread in $^{238}$U/$^{206}$Pb
ratios. This potentially improves the resolving power of a regression of the measured
isotopic ratios allowing definition of, ideally, the high-μ (radiogenic lead) and low-μ (initial
lead) end-member compositions of the data array (see Figure 2). Along with the generally
high-$n$ datasets generated by the LA-ICP-MS approach, these well-constrained regressions
can result in similar or even greater precision age determinations than those using ID data
alone. However, a caveat to this, is that lower precision data points can mask true
geological heterogeneity.




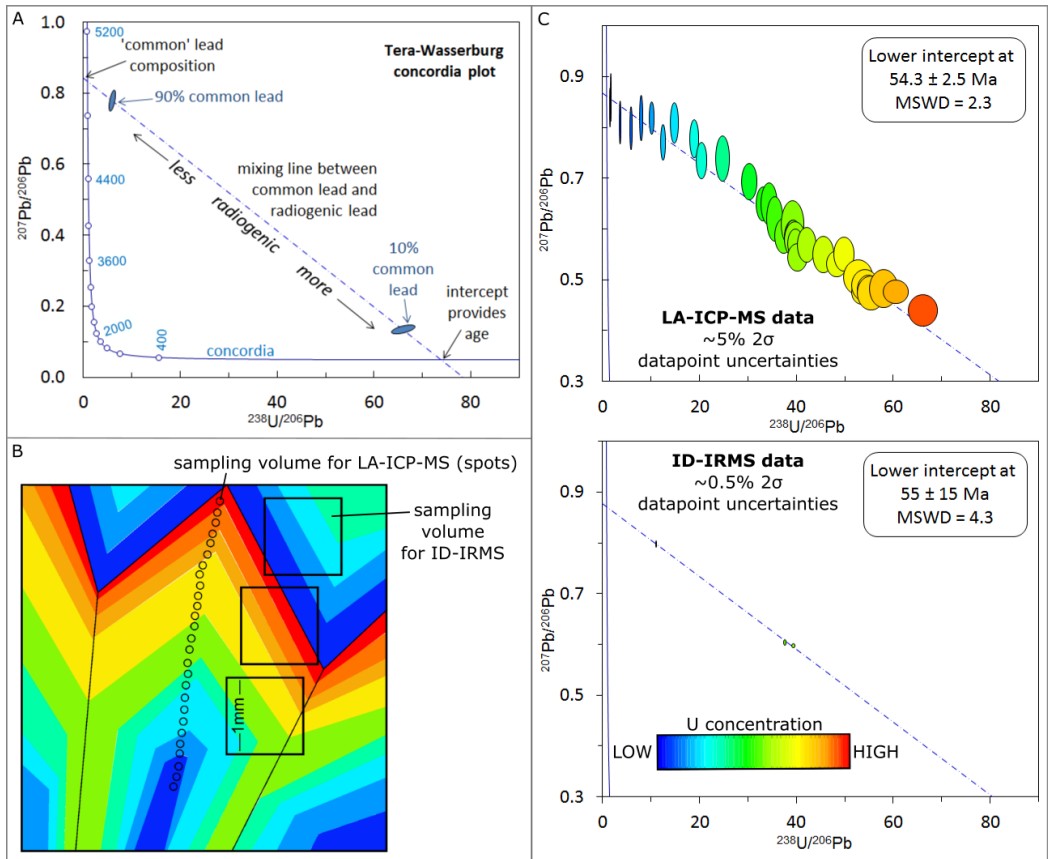


*Figure 2. A) Example Tera-Wasserburg Concordia plot demonstrating the*
*functionality of this plot for common-lead bearing U-Pb data. B) Schematic*
*model of a calcite crystal with uranium zonation indicated by the colour-scale.*
*Typical relative sample size for ID shown by the black squares, and LA-ICP-MS*
*by the circles. C) Resultant U-Pb data in Tera-Wasserburg concordia assuming*
*constant Pb concentration across the sample, and varying U concentration, for*
*LA-ICP-MS sampling and analyses versus 'bulk' sampling and ID analyses, as*
*represented by the sampling in B. The uncertainty on the datapoints is 4-6%*
*(2s) for LA-ICP-MS and ~0.5-0.7% for ID.*





When calculating an age and uncertainty from a regression/isochron, it is assumed 1) the
dataset describes a single age population whose variability or scatter is derived solely from
the analytical process, 2) each analysis represents a closed system, and 3) all analyses
share the same initial Pb isotope composition. When these assumptions are satisfied, the
MSWD should be about 1 (Mean Squared Weighted Deviation; Wendt and Carl, 1991). LA-
ICP-MS data-points generally have a lower precision than those derived by ID. These lower
precision data-points can mask scatter that exists within the level of the data-point
uncertainties. This caveat must be considered when interpreting regressed data (or
weighted means).

In other words, age interpretations rely on isochron assumptions that can only be resolved
at the level of the data-point uncertainties. More precise ID data, therefore, have better
resolution of scatter and better constrain the likelihood that a sample does not comprise a
single population. However, sampling for ID can also contribute to this scatter by analysing
larger amounts of material, with a greater chance of including altered zones or zones from
different generations. A combination of ID and image-guided LA methods can therefore
better elucidate the likely variability in any particular sample. For applications where the
best possible precision is needed (e.g. for stratigraphic constraints or charatcterisation of
potential U-Pb carbonate reference materials), a workflow involving both LA-ICP-MS dating
followed by ID on the most favourable material is the most effective. For applications where
the required precision is on the order of several percent, image-guided LA-ICP-MS without
ID is suitable.
**3.**
**4.  Identifying suitable carbonate material for dating**
4.1. $\mu$ ($^{238}$U/$^{204}$Pb) in carbonate
An 'ideal' U-Pb chronometer requires incorporation of U (the parent isotopes $^{238}$U and $^{235}$U
which decay to $^{206}$Pb and $^{207}$Pb respectively), and zero or low concentrations of initial (or
'common') Pb during its formation; this is typically expressed as the ratio of parent uranium
to initial Pb - $^{238}$U/$^{204}$Pb, or $\mu$. In addition, both the parent and daughter isotopes ideally
remain a closed system from formation until present-day. Many chronometers lack these
ideal criteria but still provide successful materials for dating: the subset of 'common-lead
bearing chronometers' comprise small to large initial lead concentrations that are of uniform
composition (e.g. titanite, apatite). The ideal criteria are generally rare in carbonates, but





many carbonate materials from a range of different geological environments are successful
common-lead bearing chronometers.

Rasbury and Cole (2009) showed that carbonates of meteoric origin have the highest μ
values, and hydrothermal varieties the lowest, with marine varieties in the middle (Figure.
3A-C). However, the recent literature on calcite dating demonstrates that with careful
characterisation and sampling, high μ domains can be found in a range of hydrothermal
and diagenetic calcite. As an example, we plot the range of μ values recorded in very low-U
(<200 ppb) and low-Pb (<20 ppb) calcite taken from basalt-hosted fractures in the Faroe
Islands (Figure. 3D). The range of mean μ values across the fifteen samples is very large
(ca. 100 to 100,000), and the range within each sample is also commonly two to four orders
of magnitude. Of the samples providing successful U-Pb isochron ages (Roberts & Walker,
2016), μ values extend to as low as ~2000.

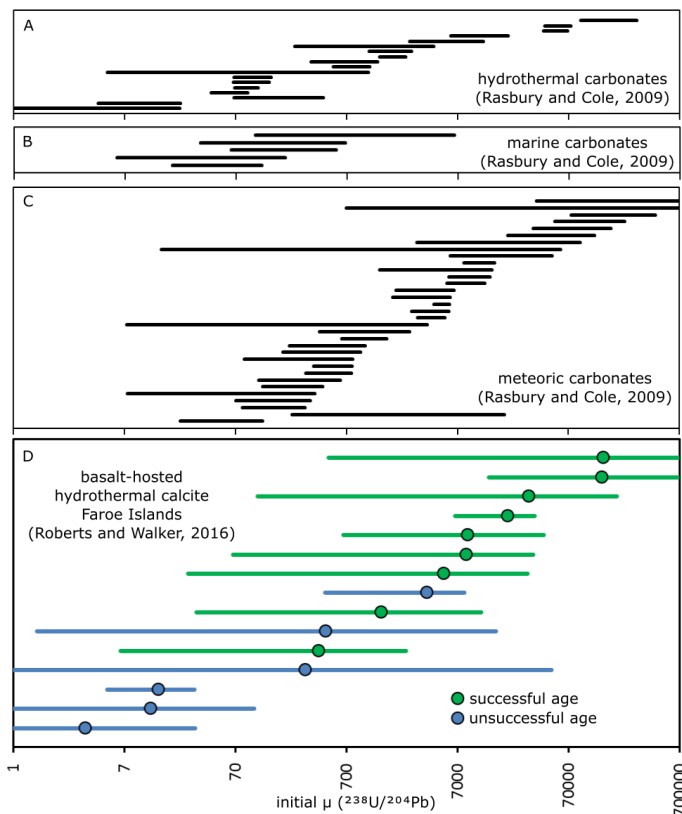




*Figure 3. A-C) Compilation of μ (²³⁸U/²⁰⁴Pb) values taken from Rasbury & Cole (2009).*
*Each bar is the range exhibited by an individual sample. These data were acquired using*
*sampling by physical separation, i.e. a dental drill. D) Compilation of μ values from basalt-*
*hosted vein-filling calcite in the Faroe Islands to highlight the range within crystals and*
*across a single region; each datapoint represents the median, and the bar represents the*
*range. These data represent laser ablation sampling.*


The amount of U needed to generate an age is dependent on two factors: (1) the age of the
material and (2) the initial μ ratio of the material. The younger a sample is, the less time
there is for the growth of radiogenic daughter Pb from parent U. With a higher μ, the ratio of
measured radiogenic Pb to common (initial) Pb will be higher, giving greater confidence and
(in general) precision and accuracy to the resulting age determination.

The effect of these factors is shown in Figure 4. Two Tera-Wasserburg plots are shown,
with isochrons for samples of different ages (100 to 10 Ma on the left, 1000 to 100 Ma on
the right). The most accurate and precise age determinations, i.e. those that can be
interpreted with most confidence, are generated when the sample comprises abundant
radiogenic lead, i.e. gets close to the lower part of the concordia curve where the
regression intercepts. Each plot shows regressions for individual samples between a
common-lead composition (~0.8) and a radiogenic end-member (with the age labelled). The
colour-coded points along each regression reflect the amount of radiogenic lead that will be
created by decay of ²³⁸U, based upon the given μ value. For example, utilising the left plot,
a sample of 15 Ma, with a μ of 10,000, will have a measured ²⁰⁷Pb/²⁰⁶Pb of ~0.4, equalling
about a 50:50 ratio between radiogenic and initial lead. To get a near concordant
measurement of this sample would require a μ value of over 200,000.  These plots
demonstrate how older samples are more amenable to dating than those young in age, at
least when regarding the abundance of radiogenic lead. The preservation of a closed
isotopic system over long time periods is what makes dating old samples (i.e. Precambrian
materials) potentially difficult.






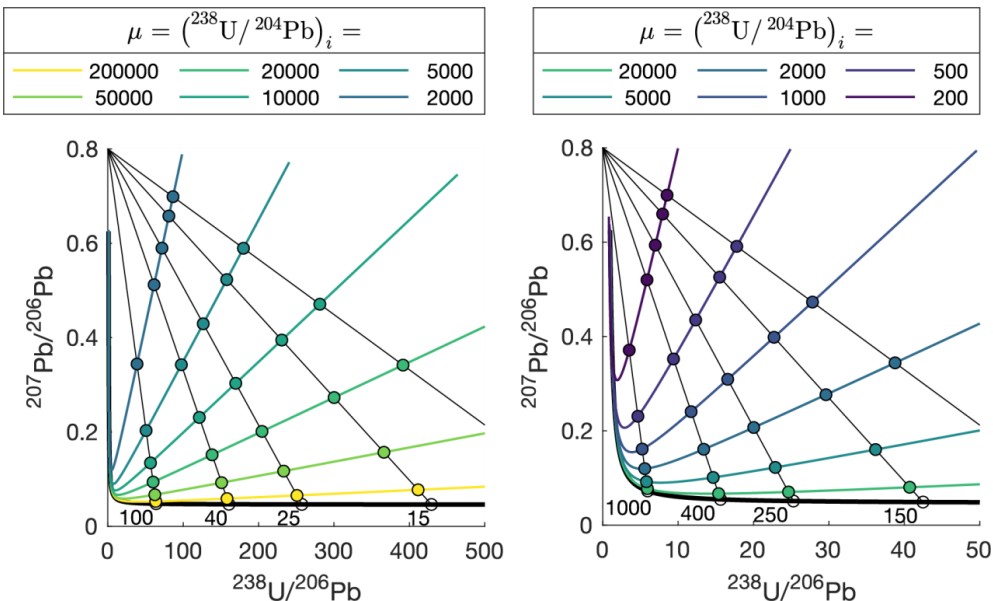

*Figure 4. Tera-Wasserburg plots showing modelled regressions for samples of different age. Colour-coded spots relate to the measured isotope composition a sample would have at a given μ value (legend above). Ages of each regression in Ma are labelled adjacent to the lower intercept with concordia.*

When absent of concordant analyses, both high μ and a significant spread in initial μ values are required to generate the most robust ages, as these will pin the isochron at the radiogenic end-member with greater confidence. Some calcite exhibits sufficiently high μ to generate concordant data (e.g. Richards et al., 1998; Roberts & Walker, 2016; Nuriel et al., 2017). Such robust ages are rare with a material that so commonly exhibits high initial lead abundances. Ages can be derived from isochrons with low amounts of radiogenic lead, i.e. those with low μ. Such isochrons can be regressed to provide lower intercept ages, but the confidence in these ages is subject to having well-behaved data conforming to a single population, requiring precise data-point uncertainties (e.g. Figure 5G). Such low μ isochrons can potentially give inaccurate lower intercept ages if the material is very young, and thus confirmation through multiple samples and/or alternative age constraints are favoured.



In Figure 5, we present a selection of 'real-world' data to highlight the potential complexity
of carbonate U-Pb data. These data from natural samples broadly range from undesirable
to most desirable from A to I, with the following notable characteristics:
A) Dominated by common lead with large data-point uncertainties (due to low count-rates)
that hamper the distinction between open-system behaviour and radiogenic ingrowth of
lead.
B) All analyses are ca. 100% common lead, with high count-rates providing a precise
measurement of the composition of this common lead.
C) Mixed and scattered data that do not fall on a single linear isochron. This is likely caused
by open system behaviour, potentially involving both addition and subtraction of parent
$^{238}$U.
D) Majority of data define a linear array with a large spread in U/Pb ratios. Some other
analyses fall on a horizontal array, suggesting they experienced open-system behaviour
(e.g., local $^{238}$U mobility).
E) Data form an apparent single linear array, but large uncertainties (due to low count-
rates) may obscure mixed ages or minor open-system behaviour.
F) Dominated by relatively radiogenic isotopic compositions, but with large data point
uncertainties due to low count-rates. The narrow range in µ leads to a large age uncertainty
from extrapolating to the lower concordia intercept. The age uncertainty would be improved
with a common lead composition estimated from contemporaneous low-µ samples of the
same suite.
G) A 'small scale isochron' (see Ring & Gerdes, 2016). There are no radiogenic isotopic
compositions to anchor the extrapolation to a lower intercept concordia date, but a tight
data array yields a realistic intercept age.
H) Dominated by radiogenic isotopic compositions, and the spread in the array provides a
precise lower intercept date; small data-point uncertainties improve ability to identify
potential outliers.
I) A precise regression due to well-behaved closed system behaviour, high count rates
giving small uncertainties, and a large spread in U/Pb ratios providing a precise estimate of
both the age and the common lead isotopic composition.

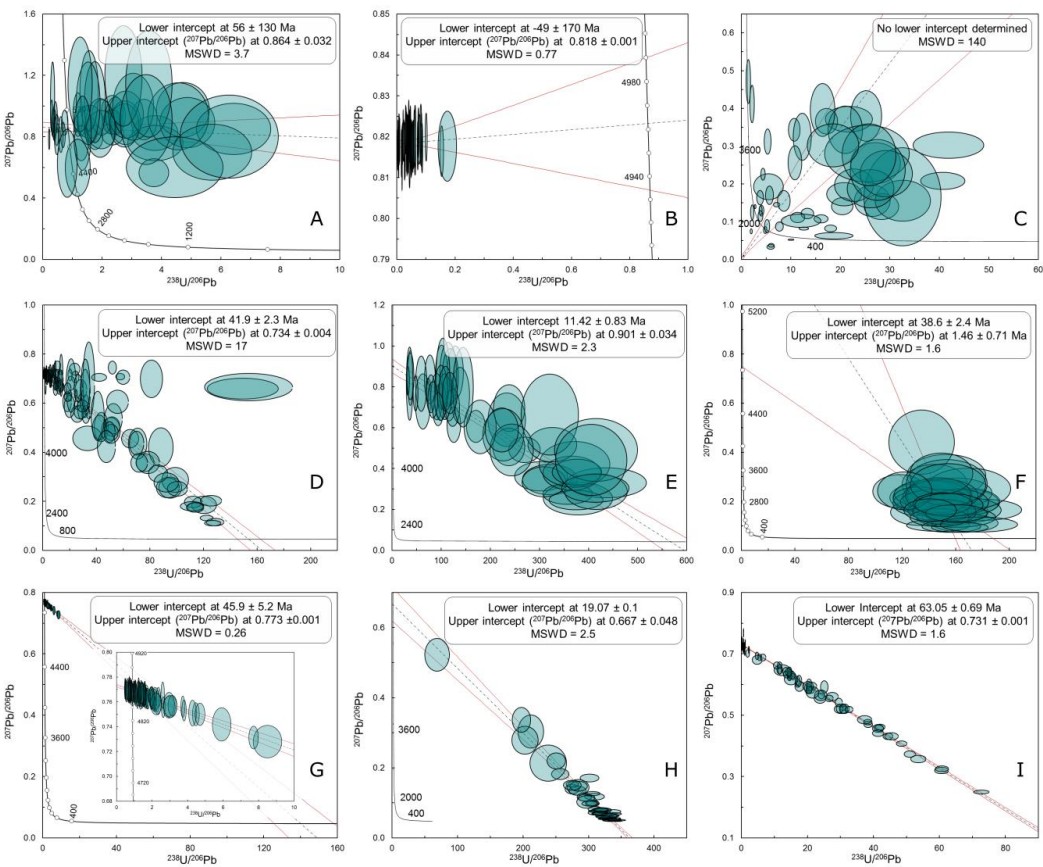

*Figure 5. Tera-Wasserburg concordia plots of natural carbonate samples from a variety of settings, with no data rejection. Lower intercept dates are quoted without propagation of systematic uncertainties. See text for explanation.*

### 4.2. U and Pb contents in carbonate

At present, there is a lack of predictive criteria that can be used in the field or in the laboratory to screen samples prior to analysis for high μ domains. Radionuclide incorporation in calcite is not well understood despite several decades of interest, primarily driven by the field of nuclear waste storage and characterisation (e.g. Langmuir, 1978; Milton & Brown, 1987; Sturchio et al., 1998; Reeder et al., 2000, 2001; Kelly et al., 2003; Weremeichik et al., 2017; Drake et al., 2018). This is because trace element incorporation in calcite does not rely on thermodynamically determined partition coefficients, but by a



large number of phenomenological variables, including: trace element availability, calcite
growth rate, temperature, pH, Eh, $pCO_2$ and the $Ca^{2+}:CO_3^{2-}$ ratio in solution, ionic size, and
U complexation. Furthermore, different trace elements can be preferentially incorporated
into structurally different growth steps and faces of growing calcite crystals (Paquette and
Reeder, 1995; Reeder, 1996).

Rasbury and Cole (2009) provided a geochronology-focused review of U and Pb in calcite,
and we note the following salient features: 1) Pb is both particle reactive and relatively
insoluble; 2) Pb is found at very low levels in most fluids (ppt-ppb), providing high Ca/Pb
ratios; 3) Pb can substitute for Ca in the crystal lattice, although the Pb cation is larger –
ionic radii of $Ca^{2+}$ and $Pb^{2+}$ in six-fold coordination are 114 and 133 pm, respectively; 4) U
exists in multiple oxidation states, and its solubility is strongly affected by Eh and pH; and 5)
both U(VI) and U(IV) have been found in calcite, but with the latter being interpreted as the
most likely and most stable form.

Points 4 and 5 above are important for understanding why and when uranium is
incorporated into calcite, and whether remobilisation is likely. Sturchio et al. (1998), using a
combination of X-ray absorption spectroscopy and X-ray microprobe fluorescence,
demonstrated that the uranium in a sample of spar calcite was in the form of U(IV), and that
U(VI) was less likely based on size and ionic structure (ionic radii of U(IV) and U(VI) in six-
fold coordination are 103 and 93 pm, respectively). Given that U(IV) is less mobile than
U(VI), this study provided important support for U-daughter geochronology. Kelly et al.
(2003) however, found that U(VI) as uranyl ($UO_2^{2+}$) was the dominant species in a natural
sample of vein calcite, which they considered to be more representative of typical low-U
material than the Sturchio sample. Drake et al. (2018) found much higher concentrations of
uranium in calcite precipitated from deep anoxic groundwater than experimental
determinations that were performed in oxic conditions, and interpreted this high uranium
uptake as due to incorporation of U(IV) and thus that the partition coefficient for U(IV) is
orders of magnitude larger than for U(VI). It is evident that more data from natural
carbonates in different settings are needed to more fully understand the controls on U and
Pb incorporation.

We have compiled uranium and lead concentration data from carbonates analysed in the
BGS laboratory over several years (Figure 6). From our data, we see that median U/Pb$_{total}$



ratios for speleothems are ~500, whereas median values for Mid-Ocean Ridge (MOR) and
continental vein calcite are 8.2 and 2.6, respectively. Note that these are total Pb contents,
and include radiogenic Pb as well as initial Pb, which causes the short linear trends that
represent individual samples. Samples in Figure 6 are mostly younger than 200 Ma, or < 4
Ma for the speleothems. The concentration data and U/Pb ratios demonstrate that
speleothems in general are much more amenable to U-Pb geochronology, which is why
they have been the main focus for this method until the last few years. Dating vein calcite,
with more variable and lower contents of U, and higher contents of Pb, has a lower chance
of success than speleothems (although it should be noted that the speleothems in general
have already been visually pre-screened during sampling).


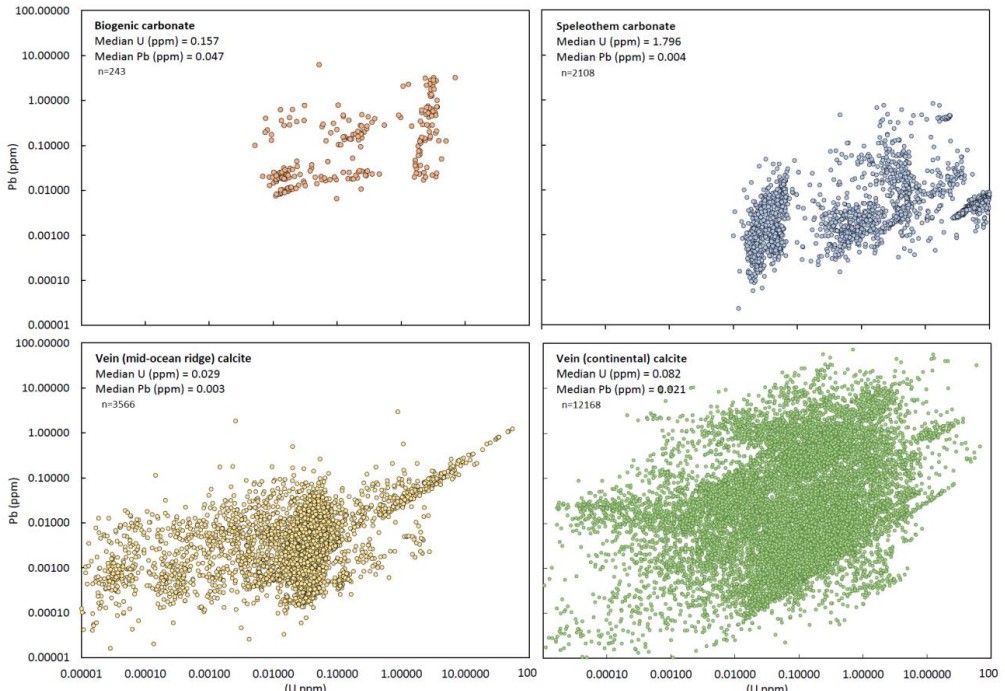


*Figure 6. Uranium and total lead contents of various carbonate materials based on a*
*compilation of laser ablation spot data from the British Geological Survey lab over several*
*years. A) Compilation of biogenic carbonates, mostly from corals; B) Speleothem*
*carbonates; C) Veins hosted within mid-ocean ridge oceanic crust; and D) veins hosted*



*within a range of lithologies from the upper continental crust, from both outcrop and*
*borehole samples.*


Ideally, a predictive framework could be constructed to aid field sampling and laboratory-
based sub-sampling of carbonate material for geochronological analyses. However, given
the large number of variables controlling U and Pb in carbonate, it is unlikely that such a
tool can be developed without measuring a large number of parameters in the
mineralising/diagenetic system. Relevant information might include the redox history of the
system. For example, oxidising fluids may mobilise U as U(VI), which is soluble in hydrous
fluids, leading to U loss during fluid-mineral interaction. Conversely, U may undergo much
higher precipitation into the mineral phase at redox fronts representing reducing conditions,
since reduced U(IV) has lower solubility. Other pertinent information for predicting success
includes the nature of the host rock and the source of the fluids. For example, if the
mineralising fluids transmit through Pb-rich units, then an undesirable enrichment in the
fluid Pb/Ca would take place, leading to lower initial $^{238}U/^{204}Pb$.

The complex nature of trace element uptake, including Pb and U, in carbonate
mineralisation is exemplified by recent studies in hydrothermal settings. Fracture
mineralisation in the crystalline basement of southern Sweden has been investigated
extensively to evaluate potential geological nuclear waste repository facilities. Several
studies have shown that most trace element concentrations vary over an order of
magnitude within calcite samples (at the thin section scale), and up to several orders of
magnitude across individual fractures (Drake et al., 2012, 2014; Maskenskaya et al., 2014;
Milodowski et al., 2018). These authors suggest that: 1) trace element chemistry does not
trace the source rock of the metals; 2) the co-variation of most trace elements implies
changing metal/Ca ratios in the fracture waters; and 3) in-situ factors affect trace element
incorporation, such as microbial activity, metal speciation, crystal habit, water type and co-
precipitation of other phases such as barite and pyrite.  Our own experience of vein-filling
fractures matches these previous studies, as shown for example by the basalt-hosted
calcite in the Faroe Islands (see Figure 9).





**5. Sample screening, imaging and petrography**
As discussed above, it is difficult to predict which carbonate samples are most suitable for
U-Pb geochronology. We therefore utilise several methods to screen material, with the aim
of limiting the time wasted on unsuitable samples, improving the quality of data that is
collected, and enhancing the overall efficacy of LA-ICP-MS U-Pb carbonate geochronology.
The purpose of sample imaging is two-fold: it provides important spatial characterisation of
U and Pb within the sample and also provides the petrographic and compositional context
to assess mineral growth mechanisms and alteration textures that are critical for linking
dates to processes.

5.1. Non-destructive techniques
A range of non-destructive imaging techniques is available for sample imaging (see Figure
7), including cathodoluminescence (CL), back-scattered electron imaging (BSE), charge-
contrast imaging (CCI), and etch-track or digital autoradiography techniques. The latter, in
particular storage-phosphor imaging plate autoradiography and direct beta-imaging
autoradiography, have been documented previously and are established techniques for
meteoric carbonates such as speleothems (e.g. Cole et al., 2003; Woodhead et al., 2012).
In carbonate minerals, CL intensity is related to trace element contents but not specifically
U concentration. CL brightness is generally ascribed to a number of emitters, with $Mn^{2+}$
being the most dominant luminescence activator and $Fe^{2+}$ being the dominant
luminescence quencher in calcite and dolomite (e.g. Machel, 1985, 2000; Savard et al.,
1995), although rare earth elements (REE) such as $Eu^{2+}$, $Eu^{3+}$, $Dy^{3+}$, $Sm^{3+}$ and $Tb^{3+}$ along
with $Pb^{2+}$ may also activate luminescence in some cases (Richter et al., 2003). Despite not
being directly related to U, the very high spatial resolution of CL is useful for identifying μm-
scale calcite crystal growth zonation and alteration, and for characterising different mineral
generations formed from different fluids (e.g. Barnaby & Rimstidt, 1989; Tullborg et al.,
2008; Milodowski et al., 2018).

BSE imaging also does not correlate directly to trace concentrations of U, but to the mean
atomic number of the mineral. It is useful as an imaging tool for characterising zonation,
alteration and growth patterns, although we note that the contrast in zonation largely
reflects variations in major element composition, and as such it is typically less sensitive





than with CL. Ukar & Laubach (2016) provide a recent review of high-spatial resolution
SEM-based imaging of vein-filling calcite mineralisation.

CCI under the SEM directly images differences in dielectric properties, which produce
charge or conductivity contrasts in the near-surface of the sample that are detected by the
secondary electron emission, and may reflect compositional variations or strain induced by
deformation (Watt et al., 2000; Robertson et al., 2005). It is an underutilised method for
geological materials, although the exact origin of charge-contrast is poorly understood.
However, it can provide useful information on crystal growth, compositional zoning and
microstructural features, and CCI has previously been applied to garnet (Cuthbert &
Buckman, 2005), feldspar (Flude et al., 2012), limestone (Buckman et al., 2016) and
biogenic calcite (Lee et al., 2008). The technique requires very clean and carefully-prepared
and polished sample surfaces because it is sensitive to surface contamination and
mechanical defects, and imaging needs to be undertaken on uncoated samples under low-
vacuum conditions.

In addition to the microscopy-based methods listed above, a lower resolution but potentially
useful technique is provided by storage-phosphor imaging-plate (IP) autoradiography using
a plastic support film coated with a photostimulated phosphor (BaFBr:Eu$^{2+}$) (Hareyama et
al., 2000). This technique records an image of the spatial distribution and intensity of total
radioactivity (from alpha, beta and gamma emitters) from a flat sample surface. In natural
geological materials, IP radiography records radioactivity from U, Th (and their radioactive
daughters), $^{87}$Rb, and $^{40}$K (Hareyama et al., 2000; Cole et al., 2003). Although U is not
specifically discriminated, it has been shown to be a useful screening tool for finding U-
bearing domains in carbonate materials (Cole et al., 2005). The method has been
particularly applied to speleothem studies where its large sample-size capabilities (up to at
least 40 cm) are beneficial. Spatial resolution is a few tens of micrometres, depending on
the pixel size of the laser scanner. However, the detection limit depends on the exposure
time of the IP in direct contact with the sample surface: routinely this is around 14-28 days
giving a detection limit of a few ppm U, which is typically higher than many carbonate
samples. Whilst this may be suitable for speleothems, which typically have higher uranium
concentrations, we do not regularly adopt the method for very low U contents in vein-filling
or diagenetic carbonates.



Fluorescence imaging has long been used in defining and characterising growth fabrics in
speleothems, although it does not specifically identify U-rich regions. This usually involves
irradiating a sliced sample with UV-light and observing the excited fluorescence emission at
a longer (visible light) wavelength, using either a standard UV microscope or digital
scanning with a UV laser system (e.g. Shopov et al., 1994; Baker et al., 1995; 2008;
Perrette et al., 2005). Fine growth detail with spatial resolutions of between 50 to 100 µm
are achievable. Speleothem fluorescence under UV at excitation wavelengths of 300-420
nm is dominated by the intrinsic fluorescence of natural high molecular weight and aromatic
organic ("humic" and "fulvic") compounds, with emission between 400-480 nm (Baker et al.,
2008). However, we have also successfully imaged speleothems (Figure 7) and other
geological materials (Field et al., 2019) by direct laser-stimulated scanning fluorescence
imaging (LSSFI) using 635 nm (red) and 450 nm (blue) wavelength excitation with 650 nm
and 520 nm low-pass wavelength filters, respectively. Although, such equipment is
principally applied to imaging of biological materials labelled with organic fluorescent dyes
(fluorochromes) (e.g. fluorescein), it is able to image variations in fluorescence originating
from organic laminae and subtle differences between carbonate minerals (calcite,
aragonite), revealing microtextural details with a resolution of about 100 micrometres.

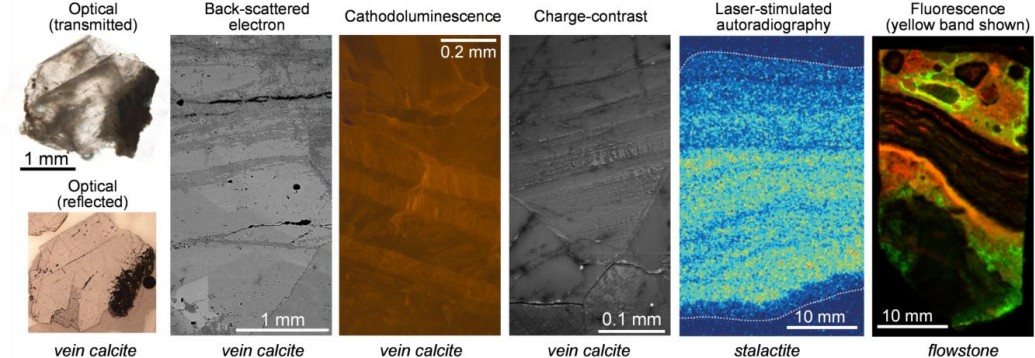



*Figure 7. Example imagery from the range of techniques used for sample screening and*
*characterisation.*






5.2. Destructive techniques
Several approaches for destructive sample screening using LA-ICP-MS are available.
These can include either systematic or non-systematic (random) spot traverses across
carbonate samples, and can include full analyses (i.e. a 30 second ablation following a pre-
ablation) or a much shorter analysis time (with or without pre-ablation). We commonly adopt
systematic traverses across samples utilising shorter ablation times but including a pre-
ablation, so as to avoid common Pb from the surface. This is a quick way to determine with
reasonable precision and accuracy whether a sample is a single age population that
represents a closed isotopic system with a suitable range in µ. For some samples, this
provides potentially useable age information that does not require any further refinement
(e.g. Figure 5H-5I). Conversely, this may provide a population of data that exhibits no
potential, i.e. dominated by common-lead (e.g. Figure 5A-5B), open-system behaviour (e.g.
Figure 5D), or mixed analyses (e.g. Figure 5C). Screening in this way allows us to analyse
several samples or sample-aliquots in a single LA-ICP-MS session, and thus identify the
material most likely to provide an accurate and precise age.

Either as an alternative to spot traverses, or subsequent to spot traverses, we use LA-ICP-
MS mapping to determine both the location and nature of U and Pb zonation in the
carbonate material. Whereas spot traverses provide rapid screening of multiple
samples/aliquots, mapping provides fairly rapid (5 x 5 mm in < 2 hours) screening across
complexly zoned samples. Different approaches can be adopted, a suite of major and trace
elements can be analysed alone, a suite of elements for age determination (i.e. Pb to U ±
Hg) can be measured, or, depending on ICP-MS instrumentation, these can be combined,
i.e. using a quadrupole ICP-MS (Drost et al., 2018) or a split-stream set-up utilising two
ICP-MS instruments (e.g. Kylander-Clark et al., 2013). As will be shown by the examples in
the subsequent sections, trace element maps are useful for directly comparing U and Pb
zonation with other trace and major elements. We have found that in primary vein-filling
calcite, U typically correlates with other trace elements, this varies between samples, but
can include V, Mn, Y, and the REEs. We can use this information to distinguish primary
zones of calcite from those that have been altered (see Section 5). Elements, or elemental
ratios such as Ba/Ca, can be used to distinguish alteration zones or secondary material
(e.g. a detrital component). For example, in meteoric carbonates, high Th is commonly
attributed to detrital matter. The production of trace element maps rapidly produces extra
information that can be related to any later age determination, facilitating the relating of the



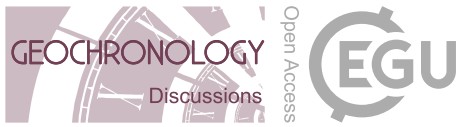

age to a specific growth event, i.e. the petrochronological approach (i.e. Kylander-Clark et
al., 2013; Engi et al., 2017).

An alternative approach is to produce maps that generate U-Pb data directly (see Section
5.5). These have obvious utility in determining suitable domains of calcite; however, for
common-lead bearing minerals they can be difficult to interpret by visual inspection. Pb-Pb
or Pb-U isotope maps can be created with ease; however, because of the inherent inclusion
of common lead, more useful is a map of common lead-corrected $^{206}Pb/^{238}U$ ages or ratios.
Common lead-corrected age maps require: 1) precise knowledge of the initial lead
composition (or upper intercept in Tera-Wasserburg space); and 2) knowledge that the
initial Pb composition is homogeneous across the mapped region, something that is not
always the case (see Section 5.4). However, with the recent advent of more advanced data
processing software, such as the Monocle plug-in for Iolite (Petrus et al., 2017), complex
age determination from maps is becoming more amenable (see Section 5.5). The caveat
with such data processing packages is that non-related domains defining a single age with
a good precision can potentially be selected with subjectivity, and without relation to actual
geological/mineralogical process. For this reason, we suggest that it is imperative that users
relate domains they have selected for U-Pb age determination to specific mineralogical
domains that can be identified independently with other means, whether these be entire
crystals, domains of crystals, growth bands, or specific veinlets. As suggested by Drost et
al. (2018), who demonstrate the method for carbonate sediments, it is also useful to
compare conventional spot ablation analyses with the map-generated dates to verify the
accuracy of the latter.

**6.  Analytical Protocol**
The LA-ICP-MS method for carbonate follows a typical sample-standard bracketing
approach using a matrix-matched reference material, i.e. as for other silicate or phosphate
minerals (e.g. zircon, monazite, titanite, rutile, apatite, allanite), with only minor
modifications. Similarly, uncertainty propagation and data reporting should follow the
community-based guidelines for zircon of Horstwood et al. (2016). Details on the LA-ICP-
MS method for carbonate adopted by three major laboratories taking a similar approach are
provided in Roberts & Walker (2016) and Drake et al. (2017) for the British Geological
Survey laboratory (Nottingham, UK); Ring & Gerdes (2016) and Methner et al. (2016) for





Goethe-Universität (Frankfurt, Germany), and Nuriel et al., (2017, 2019) for University of
California Santa Barbara (Santa Barbara, USA). There are two key points of the method we
feel are worth highlighting that differ from 'standard' methods based on silicate minerals
such as zircon. Firstly, the heterogeneous nature of the Pb isotope composition of matrix-
matched, i.e. calcite/dolomite, minerals (due to variable common Pb incorporation), means
that normalisation of the Pb-Pb isotope ratios is currently achieved using a synthetic glass
rather than a carbonate, typically NIST612 or NIST614. At present, there is no evidence to
suggest that the Pb/Pb mass bias is variable across different matrices. Secondly,
calculation of the reproducibility of the primary and secondary matrix-matched reference
materials, which is required for uncertainty propagation (Horstwood et al., 2016) and
determination of the true method accuracy and precision, is hindered by the fact that the
carbonate reference materials currently employed have U/Pb heterogeneity that is equal to
or much larger than the analytical uncertainties (Roberts et al., 2017). This means there will
typically be a significant excess variance of the reference material U/Pb isotope
measurements in any one session (including after correction for common lead), which does
not describe the reproducibility of the analytical system but instead reflects the natural
variation in the reference material. If propagated onto the sample data-point uncertainties
as a within-session excess variance as recommended for zircon in Horstwood et al (2016),
these data point uncertainties will be overestimated masking any smaller scale, real
geological scatter in the sample isochron and result in meaningless ages with erroneously
high precision. For this reason, it is suggested that calculation of the session-based
reproducibility is best estimated using a more homogenous material such as NIST glass or
zircon. However, it should be noted that through this practice results can only be compared
in a relative sense within session, or between sessions if validation materials are compiled
and used. To compare data in an absolute sense, i.e. to assign an age and total uncertainty
to a material for comparison between laboratories and/or with other methods, the
uncertainty from the primary reference material must be included to reflect the accuracy
with which the matrix-matched normalisation is known. In this way, the uncertainty of the
primary reference material constitutes a limiting uncertainty on any sample age. Improved
reference materials with less scatter around the U/Pb isochron are therefore a pre-requisite
for improving this method.



### 7. Generating U-Pb data and interpreting ages

Generating ages and relating these to geological processes requires the marriage of spatially-resolved variations in composition (elemental and isotopic) and U-Pb isotopic concentrations. In this section, we present several case studies to highlight our approach to dating vein-filling calcite, the potential applications to dating faulting and fluid-flow, and the type of material commonly encountered. First we present the standard approach, which used independent imagery and analysis to target, refine, and interpret the U-Pb analyses that are based on static spot ablations. This is the same approach as using CL imagery to help interpret zircon dates, and that can be further refined with information such as companion trace element data. A second approach (age mapping) is to use mapping tools not just to image the sample and its composition, but to extract age data from the map itself (Petrus et al., 2017; Drost et al., 2018).

#### 7.1. Image-guided dating

*Example A - Variscan-related veins in the Northumberland Basin*

Figure 8 shows U-Pb calcite data from Howick Bay in the Northumberland Basin, NE England. The mudstone succession in the bay is faulted and weakly folded, which is postulated to be a result of transpressional stress during the Variscan orogeny (De Paola et al., 2005). Syn-kinematic calcite located within fractures has the potential to date this far-field intraplate deformation (c.f. Parrish et al., 2018). Screening data from one sample, comprising randomly located spot traverses across a crystal, are presented in Figure 8C. The data yield a regression with a large array of common to radiogenic Pb compositions, but with significant scatter (MSWD = 577), indicating some alteration and open-system behaviour and/or mixed age domains. The crystal was subsequently mapped for its trace element distribution, revealing a zone of low Ca and Mg, and high Mn, REEs and Pb. This zone can be seen optically, and is interpreted as a zone of alteration. Further U-Pb spot analyses were placed in a domain away from this feature that exhibits high U, with the data yielding a more precise regression with an age of 287 ± 14 Ma (MSWD = 2.5). This example highlights the use of trace element mapping to locate regions of highest U, to assist and refine U-Pb analyses, and shows the potential for dating calcite veins into the Palaeozoic.

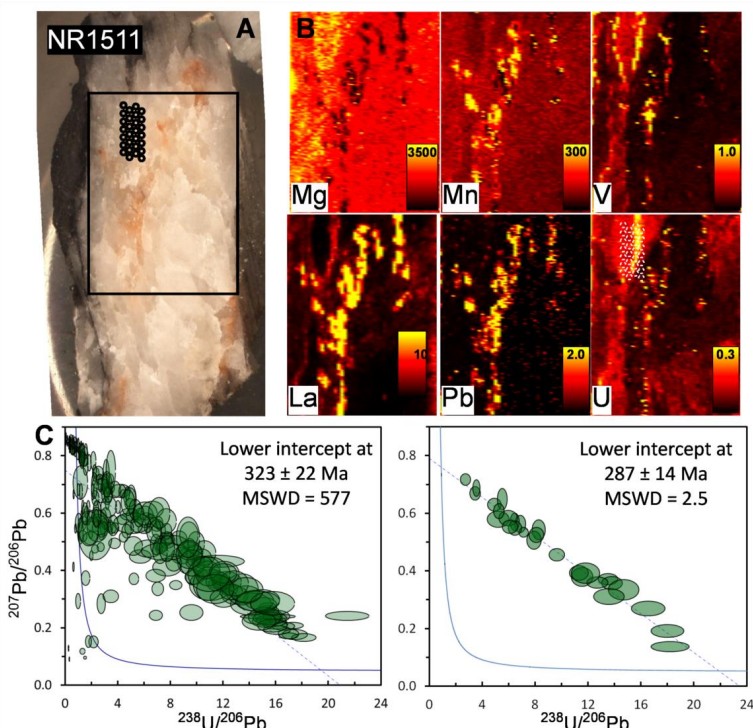

650

*Figure 8. A) Photomicrograph of sample NR1511 showing position of mapped area and*

*ablation spots (in black); B) Trace element maps generated with LA-ICP-MS using line*

*rasters, the scales are in ppm and white spots show the ablation locations; C) Tera-*

*Wasserburg concordia of U-Pb data from this sample based on screening spot traverses;*

*D) Tera-Wasserburg concordia of U-Pb spot data placed using constraints from map data,*

*i.e. over the unaltered high U region.*



*Example B - Faroe Island vein mineralisation*
The aim of most studies is to date primary calcite formation rather than subsequent
secondary alteration, particularly when dating syn-kinematic calcite for constraining the
timing of fault slip (e.g. Roberts & Walker, 2016; Ring & Gerdes, 2016; Goodfellow et al.,
2017; Hansman et al., 2017; Nuriel et al., 2017, 2019; Parrish et al., 2018; Holdsworth et
al., 2019; Smeraglia et al., 2018). Trace element mapping is a useful tool to assist with
identification of growth zoning, particularly on the scale of mm- to cm-sized chips. Using
standard LA-ICP-MS protocols for trace element determination, with standard-sample



bracketing routines, a 5 x 5 mm region can be mapped in less than two hours. As discussed
previously, depending on the analytical set-up, this trace element mapping can be
conducted alongside U-Pb isotope mapping.

Figure 9 shows selected results from dating of syn-kinematic crack-seal-slip calcite
mineralisation from basalt-hosted faults of the Faroe Islands (Roberts & Walker, 2016).
Sample A is from a vein exhibiting a zeolite-calcite-zeolite mineral paragenesis. The calcite
exhibits distinct syntaxial growth zoning towards the centre of the vein. Trace element
mapping reveals large variation in trace element contents in the direction of growth,
interpretable as changing metal/Ca ratios in the mineralising fluid (e.g. Drake et al., 2014).
The trace element zonation clearly follows the optically visible growth zonation, indicating
its primary nature. Uranium increases steadily to the maximum concentrations observed,
then abruptly drops to very low abundances. The U-Pb data define a well-behaved isochron
(low scatter with large spread in U/Pb ratios), determined from spots placed on and near
the high U region, and yields a lower intercept U-Pb age of 45 ± 2 Ma (MWD = 1.09).

Sample B is from a large dilational jog (up to 1 m wide) that is filled with zeolite-calcite-
zeolite mineralisation, including calcite crystals up to 10 cm long. The mapped grain is
composed of calcite with a later rim of zeolite. Trace element mapping reveals a strong
correlation between most elements, again, representing the primary growth zonation. High
Mn and V 'fingers' intersect the growth zonation, and are visible optically. We interpret
these as pathways of secondary alteration. Given that the vein exhibits vuggy textures, it is
possible that fluids have precipitated or altered the original calcite much later than the
original fault slip. U-Pb analyses of the primary calcite in this sample reveal fairly radiogenic
Pb compositions, although with large datapoint uncertainties owing to the low U
concentrations, with a lower intercept U-Pb age of 37 ± 2 Ma (MSWD = 2.4; anchored
upper intercept based on other samples on this study at 0.89 ± 0.02). Trace element
mapping allows us to visualise and fingerprint these alteration zones, and avoid or remove
them from analyses used for dating. A benefit to this approach is that the maps can then be
used to estimate the trace metal contents of the mineralising fluids, which in turn provides
information about rock-water interaction and the redox conditions for example. These maps
also demonstrate that no measurable diffusion of trace elements across the calcite crystals
has occurred over a significant time span, as the distribution is interpreted as a primary
feature.




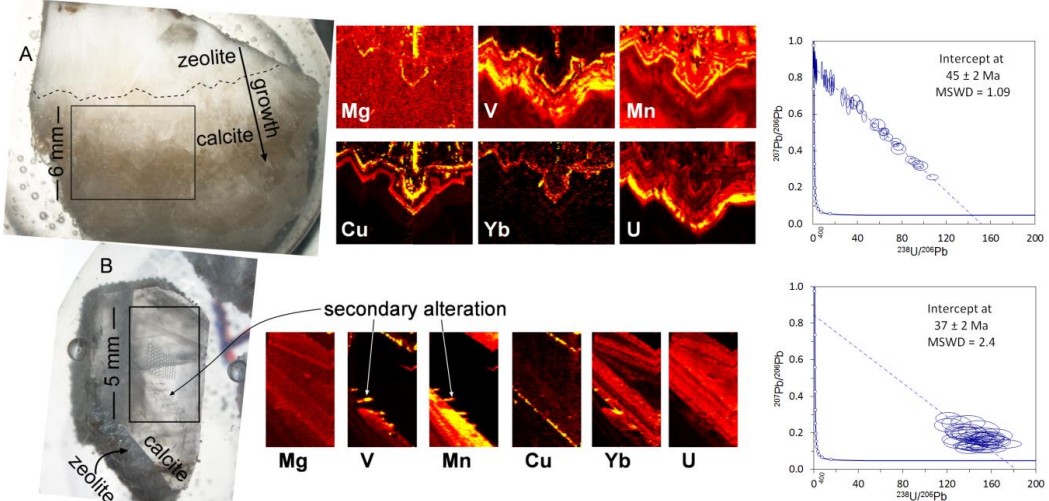



*Figure 9. A) Sample microphotographs of epoxy-resin mounted zeolite-calcite vein*

*samples, TJN-0-1 and TJN-6-1, mapped regions shown in black; B) Trace element maps*

*generated with LA-ICP-MS using line rasters; C) Tera-Wasserburg concordia of U-Pb data*

*from each sample (from Roberts and Walker, 2016).*

708

709

*Example C - Sellafield fracture mineralisation*

Figure 10 shows results for fracture mineralisation from Sellafield, NW England. This

location was previously investigated (1990-1997) as a potential nuclear waste repository

site, and has thus been extensively studied in terms of its structure, stratigraphy,

geochemistry and palaeo- to present-day hydrogeology (Michie & Bowden, 1994; Akhurst

et al., 1997; Baker et al., 1997; Heathcote and Michie, 2004; Bath et al., 2006; Milodowski

et al., 1998, 2018). Deep (up to 2 km) site-investigation boreholes revealed a complex

sequence of fracture mineralisation within Ordovician greenschist-facies metamorphic

basement rocks overlain by Carboniferous Limestone and Permo-Triassic sedimentary

strata (Milodowski et al., 1998). Presented here are data from one mineralised zone that

show the potential for U-Pb dating of such material.


Sample 833 is an example of euhedral calcite crystals lining open fractures and

representing the latest mineralisation, and which are very closely-associated with the





present-day fracture-controlled groundwater system (generation ME9 of Milodowski et al.,
1998; 2018). The sample has U and Pb concentrations of 0.3—30 ppm and 0.1—3.6 ppm,
respectively. U-Pb LA-ICP-MS spot analyses were placed in a single crystal which was
optically continuous; the data yielded an age of 6.98 ± 0.43 Ma (MSWD = 15). The dated
crystal was subsequently mapped for its U, Th and Pb elemental distribution using LA-ICP-
MS. The map shows zoning of U, Th and Pb that is interpreted as growth zoning during
primary calcite growth. Pb is distributed similarly, but with high concentrations along narrow
veins that are discordant to the primary growth zoning; these are interpreted as alteration
pathways where Pb-bearing fluids have invaded the crystals. Crystals were also imaged
using charge-contrast imaging (CCI), which highlights structural imperfections in the calcite
crystals. The same veinlets that have elevated Pb concentrations are imaged as cracks and
disturbances to the growth zoning. Since the spots that lie on the alteration pathways have
high Pb counts, the age data were culled based on Pb concentration (>500 ppb Pb
removed). This approach reduced the scatter in the regression, presumably removing
components with slightly different common lead compositions, giving a more precise age of
6.44 ± 0.26 Ma (MSWD = 2.8). These data from Sellafield demonstrate the potential utility
of imaging techniques such as CCI and trace element mapping to discriminate primary
growth domains from those that are altered at the micro-scale (<100 μm), and refinement of
scattered analyses into those that are interpretable as a single population.

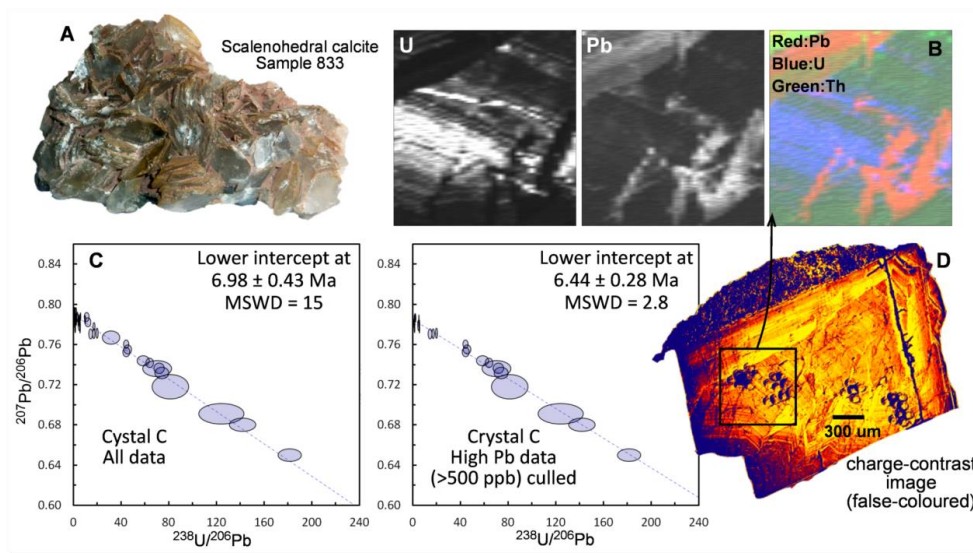






*Figure 10. A) Photograph of sample 833; B) Trace element maps using LA-ICP-MS based*
*on line rasters; C) Tera-Wasserburg concordia of U-Pb data before and after refining the*
*data; D) False-colour charge-contrast image of the dated crystal, showing the mapped*
*region in black.*

7.2. Age mapping of vein-fill carbonates
We have demonstrated that elemental mapping data are useful for refining and interpreting
U-Pb isotopic data. For example, in Example B above, we manually located the spots in a
high U zone, and in Example C, we manually removed the data with high Pb
concentrations. An alternative approach to using the maps to 'manually' locate spots or
refine spot data, is to generate a combined elemental and U-Pb isotopic 2D dataset (i.e.
map); the benefit of this method is that software tools can be used to both discriminate
specific isotopic data based upon chosen criteria, and also to show regions within these
pooled datasets that have similar compositional characteristics. Iolite (Paton et al., 2011) is
one of the most commonly used data reduction tools for both U-Pb isotopic data (Paton et
al., 2010), and for generation of elemental 2D maps. Monocle is a software plug-in for Iolite
that allows the user to generate maps of isotopic and elemental data (Petrus et al., 2017),
and to define and extract regions of pooled compositional data, including those used for
age calculations. Drost et al. (2018) demonstrated the efficacy of the software for dating
carbonate sediments, whereby features such as bioclasts and detrital components are
removed. For a detailed explanation of the protocol, see Drost et al. (2018). In brief, each
pixel of the elemental and isotope ratio maps corresponds to one duty cycle of the ICP-MS.
First, pixels are removed, using user-defined selection criteria that are believed to be
related to alteration, secondary material, or a younger or older carbonate generation. This is
usually conducted after an initial inspection of the mapping data combined with prior
imaging and petrography; however, the screening can also employ an iterative approach
after generation of initial U-Pb isochrons. After this screening/filtering, the remaining data
are pooled into a number of pseudo-analyses (each corresponding to the same number of
pixels) based on a suitable isotope ratio, such as $^{238}U/^{208}Pb$ or $^{235}U/^{207}Pb$. The pooling is
achieved using an empirical cumulative distribution function ECDF) to maximise the spread
in U/Pb ratios, and an appropriate number of pixels to produce a reasonable population of
data, for example twenty to forty data-points. Here, we present examples of this approach
applied to vein-filling calcite.






*Example D – BH11*
This example is of a fine-scale vein cross-cutting a sedimentary host rock; the objective is
to use Monocle-based criteria to discriminate the vein from the host rock and determine a
robust age. Only two criteria of filtering were needed to distinguish the vein from the host:
Mg of less than 5000 ppm, and Th of less than 0.1 ppm.  The remaining data were pooled
using $^{238}U/^{208}Pb$ ratios into 26 analyses, and yielded a robust lower intercept $^{238}U/^{206}Pb$
date of 53.95 ± 0.36 Ma, with an MSWD of 1.0 (Figure. 11). This sample was previously
dated using conventional spot analyses located within the vein at 53.51 ± 0.39 Ma (MSWD
= 2.0; Beaudoin et al., 2018). These dates, quoted without propagation of systematic
uncertainties, show good agreement between two different labs using different
instrumentation and data reduction methods.

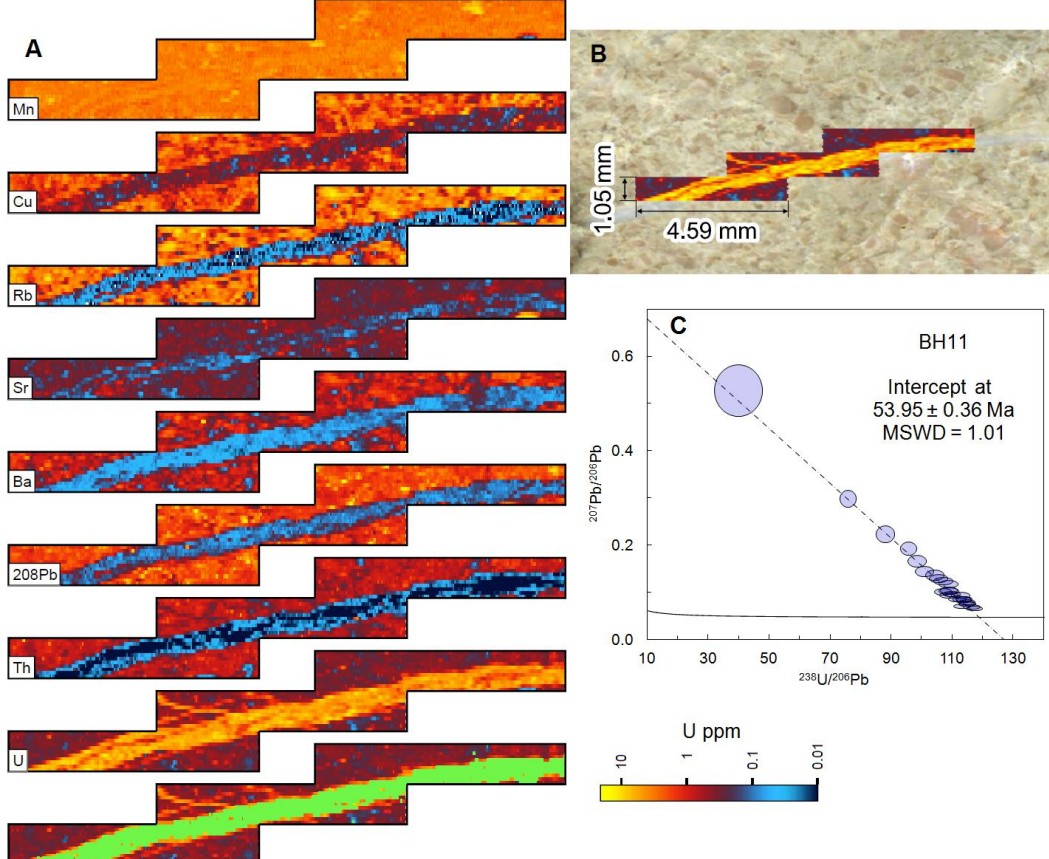




*Figure 11. Image-based dating (Monocle plug-in for Iolite) of sample BH11. A) Trace*
*element maps of the analysed region, the final map shows the region of interest selected*
*for the U-Pb date highlighted in green; B) Photomicrograph of the sample surface showing*
*the mapped region as a U map; C) Tera-Wasserburg concordia of U-Pb data after pooling*
*and filtering using the Monocle plug-in (see text for description).*


*Example E – BM18*
BM18 is another example of a vein cross-cutting a sedimentary host-rock. This time, there
is clear zonation of the vein (Figure. 12). Since it is a syntaxial vein (crystals growing from
the wall rock to the centre), this zonation probably represents changing fluid chemistry as
the calcite crystals were precipitating. However, it could represent multiple generations of
calcite precipitation. Criteria were selected for filtering of the data to highlight the outer
regions of the vein; Rb < 0.05 ppm, Th < 0.01 ppm, and Sr < 400 ppm. The U-Pb data were
then filtered to remove data with low U and Pb signals, since no initial rejection of data
based on detection limit was conducted using this data reduction method; criteria for
acceptance were $^{238}U$ > 500 cps, and $^{207}Pb/^{206}Pb$ < 1.5. The remaining data produce a
robust isochron with a lower intercept date of 61.0 ± 1.7 Ma (MSWD = 1.11; 21 pooled
analyses). This date overlaps that previously obtained using spot analyses that were
derived from the entire width of the vein (59.5 ± 1.7 Ma; Beaudoin et al., 2018).

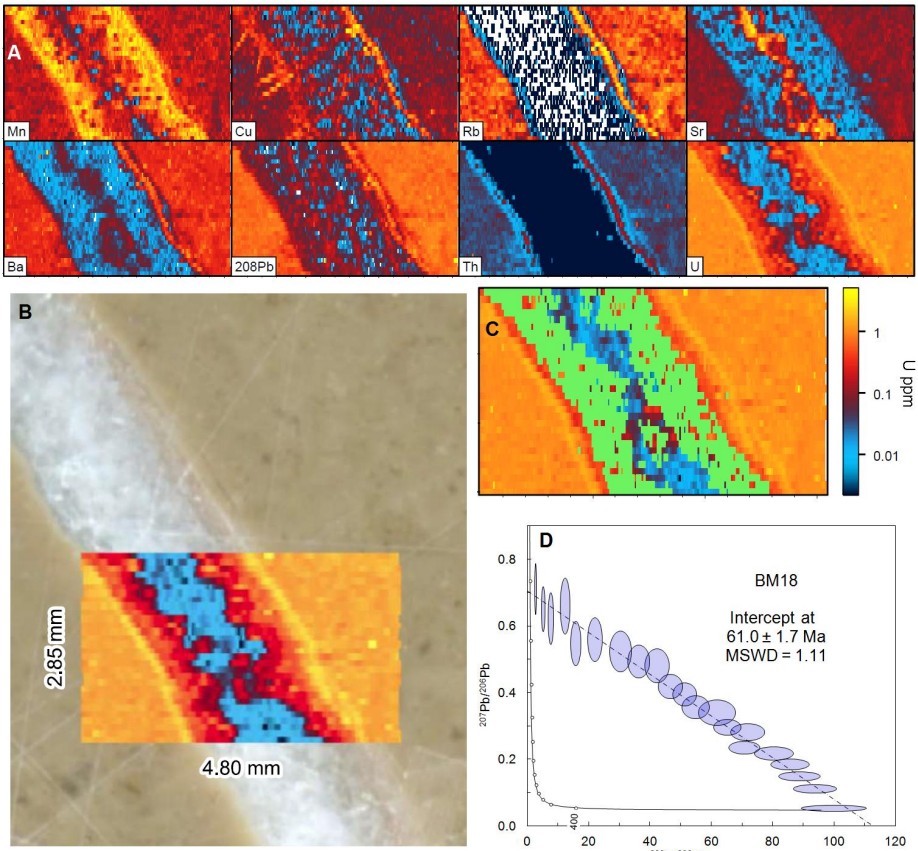


*Figure 12. Image-based dating (Monocle plug-in for Iolite) of sample BM18. A) Trace*

*element maps of the analysed region; B) Photomicrograph of sample surface showing*

*mapped region as U map; C) U map showing the region of interest selected for the U-Pb*

*date in green; D) Tera-Wasserburg concordia of U-Pb data after pooling and filtering using*

*the Monocle plug-in (see text for description).*


*Example F – NR1511*
The third example of image-based dating is from a complex vein already described in
section 5.1 above (Example A). This vein features visible textures and chemistry associated
with alteration (Figure. 13). The mapped region is entirely within the vein (no host rock).
High concentrations in several elements (e.g. Cu, Ba, Rb, Sr, Ba and Pb) reflect veinlets
that can be seen optically as a yellow altered region. The remaining portion of the vein
varies in U content, which likely represents chemical zonation across the coarse sparry
calcite growth. A fairly robust isochron (MSWD = 1.9) was obtained after filtering of the data





for the clearly altered regions, cleaning up the U-Pb data to remove low U and Pb signals,
and pooling the data based on $^{207}Pb/^{235}U$. The criteria for acceptance were: Cu < 0.2 ppm,
Ba < 10 ppm, Rb < 0.01 ppm, and $^{238}U$ < 10000 cps (for removal of alteration), and $^{238}U$ >
500 cps, $^{207}Pb/^{206}Pb$ > 0.15 < 1.5, and $^{206}Pb/^{208}Pb$ > 0.1 < 10 (for 'cleaning up' the U-Pb
data). These data yielded a date of 283 ± 9.7 Ma, which overlaps that obtained from spot
analyses and manual location of the spot data based on prior LA-ICP-MS mapping (287 ±
14 Ma; see Figure 8D), but with an improvement in the precision (4.9 to 3.4 %).

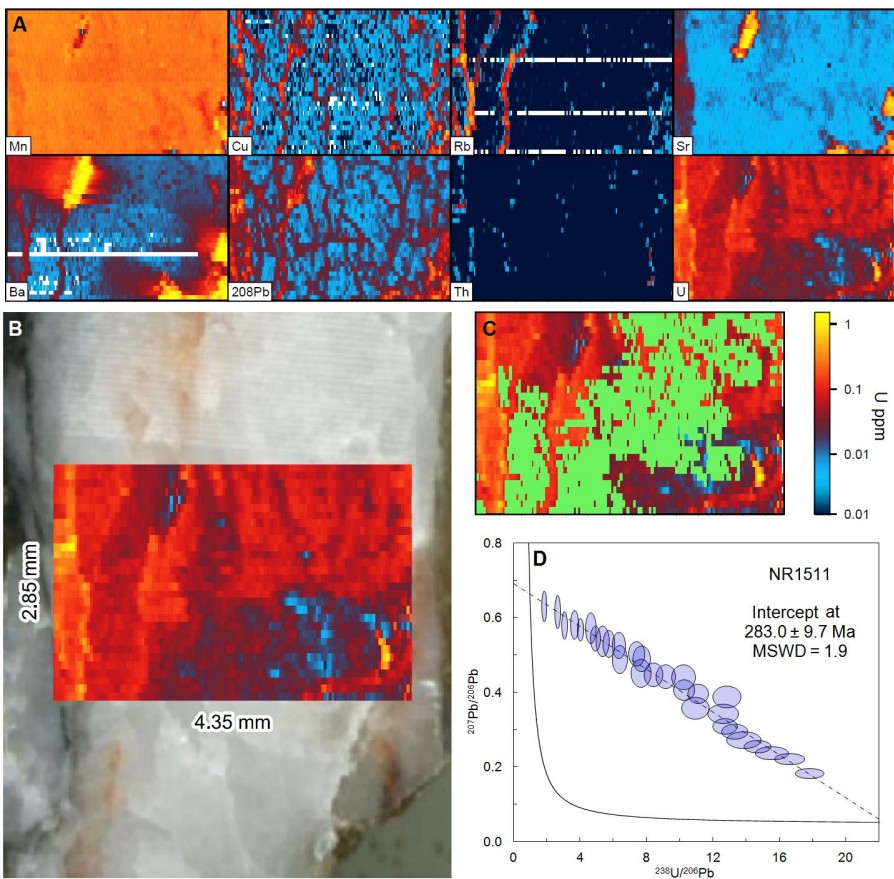


*Figure 13. Image-based dating (Monocle plug-in for Iolite) of sample NR1511. A) Trace*
*element maps of the analysed region; B) Photomicrograph of sample surface showing*
*mapped region as U map; C) U map showing the region of interest selected for the U-Pb*
*date in green; D) Tera-Wasserburg concordia of U-Pb data after pooling and filtering using*
*the Monocle plug-in (see text for description).*





**8. Isotopic composition of common lead**

Carbonates nearly always take up some amount of lead during their formation, referred to as 'common' or initial lead. Contamination during handling (i.e. during cutting and polishing) or from recent exposure to the environment will have a modern isotopic composition of common lead, i.e. approximating the Stacey & Kramers (1975) model for terrestrial lead composition at present-day, roughly $^{207}Pb/^{206}Pb = 0.84$. Distinguishing between such contamination and the common lead incorporated during formation can be difficult. Well behaved U-Pb isotopic systematics in a carbonate sample should yield a single mixing line between the common and radiogenic end-members, and ideally will have enough spread in U/Pb ratios to yield a precise regression with low uncertainties at both the lower (radiogenic lead) and upper (common lead) intercepts. However, many samples will exhibit a lack of spread in U/Pb ratios, or will be dominated by radiogenic compositions (e.g. Figure 5F). Although a best-fit line may be calculated for such data, the slope, and thus age, may be inaccurate. Thus, it is useful for such samples to have an estimation of the common lead composition through other means, such as from nearby cogenetic samples formed at the same age, or from different minerals also believed to have been formed at the same age.

For some mineral chronometers, such as the phosphate mineral monazite, it is common to use an estimate of the common lead composition based on the Stacey and Kramers (1975) model (e.g. Palin et al., 2013; Regis et al., 2016). In our experience, this is an acceptable approach because from a number of different studies, we find that the common lead composition determined from other minerals (i.e. feldspar, biotite, apatite) overlaps the Stacey and Kramers (1975) composition (e.g. Stübner et al., 2014; Warren et al., 2014). For carbonate however, we find this is not always such a suitable approach. Our experience from hydrothermal carbonate in particular, is that common lead compositions are often more radiogenic (lower $^{207}Pb/^{206}Pb$ ratios) than the terrestrial lead model (Stacey and Kramers, 1975) for age of carbonate crystallisation. This can occur if the carbonate has incorporated lead during its formation that is derived from ancient sources. Figure 14 shows a compilation of common lead intercepts from a number of studies of fracture-filling calcite (compilation of BGS laboratory data). The data represent different host lithologies, different ages (dominated by Cretaceous to Miocene), and different geological regions. It is clear that for many samples in this compilation, anchoring at a value close to the terrestrial lead model composition for Phanerozoic ages, i.e. $^{207}Pb/^{206}Pb \sim 0.84$, will lead to calculated ages older than the true age due to steepening of the regression. The importance of the



common lead composition in providing constraints on a calculated age will depend on the
amount of measured radiogenic lead in a given sample; samples dominated by common
lead and lacking in radiogenic lead will need a well defined array to produce a confident
lower intercept. We find that within individual vein samples, the apparent composition of the
common lead end-member can vary, limiting the precision of the regression and derived
age. For speleothems, Woodhead et al. (2012) demonstrate that most samples analysed in
their lab yield common lead compositions overlapping Stacy and Kramers (1975), and thus
their ages are largely insensitive to the common lead compositions. This likely reflects the
fact that they are precipitated from meteoric water that reflects the regional upper crustal Pb
composition. Although, they add the caveat that samples with $^{238}U/^{206}Pb$ below 1300
(equivalent to μ ≈ 20,000), have large inaccuracies.

The highly radiogenic initial lead values ($^{207}Pb/^{206}Pb$ < ~0.75) recorded in our compilation
are mostly from two settings, young fractures in Proterozoic crystalline crust of Sweden (n=
10 of 104), and young fractures in the Bighorn Basin that overlies Archaean basement (n=
24 of 104). In both cases, the whole-rock Pb, although ancient, is not radiogenic enough to
produce the measured values. Instead, leaching of lead from uraniferous minerals is
required, e.g. titanite, allanite, monazite and zircon as a causative mechanism.


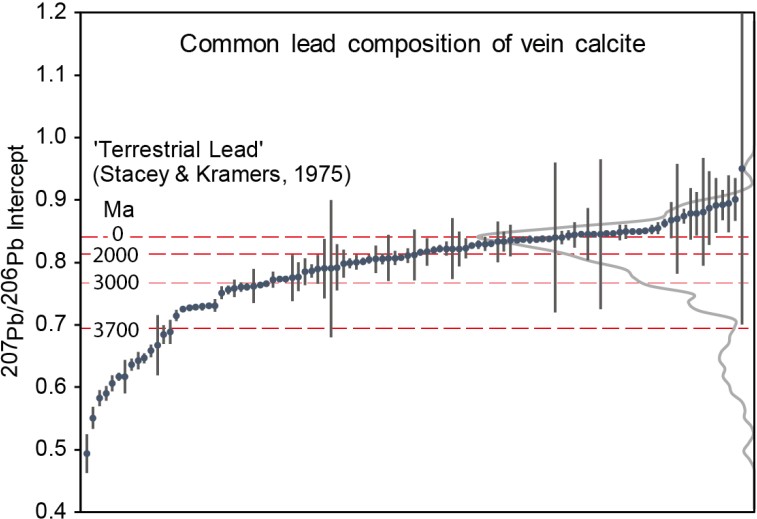


*Figure 14. Compilation of upper intercept $^{207}Pb/^{206}Pb$ compositions from vein-filling calcite*
*from samples dated in the British Geological Survey laboratory (n=104). The grey curve is a*



*Kernel Density Estimate showing the distribution of mean compositions. The red bars show*
*the two-stage Stacey and Kramers (1975) compositions of terrestrial lead at 0, 2000, 2000*
*and 3700 Ma. Samples with very large uncertainties in the $^{207}Pb/^{206}Pb$ composition are*
*those with very low Pb count-rates.*


*Example G - Moab fault*
Figure 15 shows results from a sample taken from the Moab fault in southeast Utah, USA.
The sample presented here (CHJ15-KH08) is collected from the Courthouse Junction fault
segment intersection. This locality has a complex, multi-phase deformation history
(Davatzes et al., 2005; Johansen et al., 2005) associated with multiple episodes of
mineralization and a range of diagenetic fluids (Chan et al., 2000; Eichhubl et al., 2009;
Bergman et al., 2013; Hodson et al., 2016).

U-Pb data were obtained from different sections of the vein material formed along different
orientations (See Figure 15). The data exhibit a high level of common/initial lead, with
limited spread in radiogenic lead contents, but still forming a scattered regression to a lower
intercept value. Using different colours to discriminate different sections of vein, it is clear
that they have subtly different initial lead compositions, as indicated by the upper intercept
($^{207}Pb/^{206}Pb$ value) of the data arrays. These lead compositions are different from that
predicted by the Stacey & Kramers (1975) terrestrial composition, which we find is a
common feature of many vein-filling carbonates. This is likely due to the hydrothermal fluids
that are precipitating the carbonate comprising unsupported radiogenic lead components
derived from leaching of older uraniferous minerals or rocks.

The existence of variable Pb compositions on small length-scales (<1 mm) means that
careful attention is required to interpret complex data. However, the spatial resolution of LA-
ICP-MS means that these details can potentially be teased out. This case study also shows
the potential of the method for measuring veinlets that are only ~150 μm wide (see Figure
15), a task that would be difficult for ID analyses.



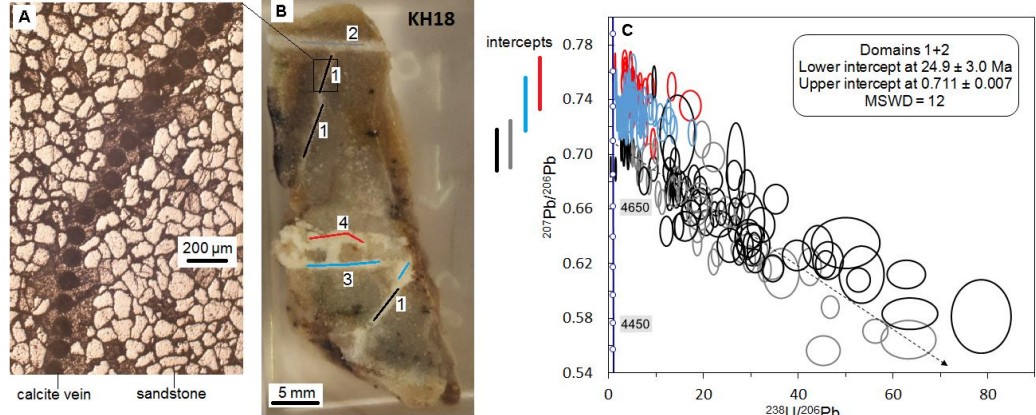

*Figure 15. U-Pb data from a series of calcite veins (sample KH18) along the Moab Fault at*

*Courthouse Junction, Utah. A) Reflected light image of a region of veining showing the 100*

*µm spots; B) Photomicrograph of the dated sample, with different dated domains of veining*

*shown by blue, red, black and grey lines; C) Tera-Wasserburg plot with U-Pb spot data*

*colour-coded to match the different domains. The bars on the left show the variable*

*$^{207}Pb/^{206}Pb$ upper intercept values for each domain.*

In summary, vein-filling, diagenetic and hydrothermal carbonates often do not exhibit Stacy

& Kramers (1975) model Pb compositions for their assumed age, but typically yield more

radiogenic compositions. This means that regressions anchored with assumed common

lead compositions are susceptible to inaccuracy. Mixed common lead compositions in

samples hampers derivation of single age regressions, implying multiple fluid sources.

Mixed ages and atypical lead compositions can also make age mapping problematic.

**9. Dating young material – dealing with disequilibria**

As described in Section 3, the younger the age of the sample analysed, the lower the

potential for precise and accurate age determination due to the lack of radiogenic ingrowth

of lead. However, young carbonates are a high priority in many applications, because they

can date events more relevant to the Earth system at present, and because U-Pb can

extend the age range of sample suites or study areas where U-Th age dating is also

feasible. For example, records of environmental change in deep time require the dating of

speleothems that are older than 500 ka (see Woodhead et al., 2012, 2019), and dating of





veins that record seismic cycles extending beyond 500 ka (see Uysal et al., 2011; Williams
et al., 2017) can provide constraints on earthquakes and other hazards associated with
subsurface fractures. These particular applications are likely to require high levels of
precision, i.e. for the Quaternary, of much less than ± 100 ka, and potentially even less than
± 10 ka or < 1000 years for the Holocene. Achieving such precision requires very high U to
achieve abundant radiogenic lead and higher μ values (see Figure 4).

A major issue for accurate dating of young samples (i.e. <10 Ma) is the potential effect of
initial daughter isotope disequilibrium within the uranium decay chains. The simplest form of
the U-Pb and Pb-Pb age equations, often used for older samples, assume that all long-lived
daughter isotopes in the U decay chain are initially present in secular equilibrium. Both the
U decay series contain long-lived daughter isotopes, including $^{234}$U ($t_{1/2}$ = 245 ka), $^{230}$Th ($t_{1/2}$ =
76 ka), and $^{226}$Ra ($t_{1/2}$ = 1.6 ka) in the $^{238}$U decay chain, and $^{231}$Pa ($t_{1/2}$ = 34 ka) in the $^{235}$U
decay chain. Of these, $^{234}$U has the longest half life and therefore the largest potential
effect on U-Pb dates. The excess initial $^{234}$U often observed in natural waters will lead to
generation of unsupported $^{206}$Pb. If uncorrected, excess initial $^{234}$U produces overestimated
$^{238}$U/$^{206}$Pb and lower intercept dates. An excess of the other intermediate daughter
products, like $^{230}$Th, relative to secular equilibrium will bias the age with a smaller
magnitude but in the same direction, whereas a deficit will result in dates that are too
young.

Carbonates are commonly precipitated from fluids containing $^{234}$U/$^{238}$U out of secular
equilibrium. Thus, this initial disequilibrium must be considered in any age determination.
Age corrections for initial U daughter deficits are at maximum ~1.44 times the half life of the
daughter isotope for zero initial abundance. But for initial excesses, the age difference can
be many times larger (see Figure 17). For most older samples dated by U-Pb, the effect of
disequilibrium is deemed to be insignificant compared to larger measurement uncertainties.
For this reason, initial disequilibrium has thus far not been mentioned in any publication
concerning LA-ICP-MS U-Pb dating except for those dealing with young speleothems (e.g.
Hopley et al., 2019). However, here we demonstrate that initial disequilibrium may be a very
significant cause of uncertainty for carbonates precipitated from groundwater and other
crustal fluids, and not just for very young (<1 Ma) samples.



In young samples, particularly those within the range of U-Th geochronology (<600 ka), the
initial $^{234}U/^{238}U$ ratio ($^{234}U/^{238}U_0$) can be estimated based on the combination of the present-
day measured $^{234}U/^{238}U$ ($^{234}U/^{238}U_{now}$), and either the measured $^{230}Th/^{238}U$ or the estimated
date of formation. The robustness of this estimate is highly dependent on the precision and
accuracy at which the isotope ratio(s) can be measured (the atom ratio is very small,
making high precision measurement >1‰ difficult). In addition, if the offset between
$^{234}U/^{238}U_{now}$ and secular equilibrium is small, then the measurement may overlap secular
equilibrium within uncertainty. For this reason, the highest precision possible is a necessary
target for any disequilibrium correction measurement.

For older samples (i.e. those older than about four times the half-life of $^{234}U$), and/or those
with only a small degree of initial disequilibrium, $^{234}U/^{238}U_{now}$ is likely to have reached
secular equilibrium. This means that $^{234}U/^{238}U_0$ cannot be estimated from the measured
data alone. One approach to alleviate this problem is to take known initial ratios from
younger samples (<600 ka) formed in approximately the same geologic setting, and apply
these corrections to the older samples from the same setting (e.g. Woodhead et al., 2006,
2019). This approach is only applicable if the geological environment is well known and the
hydrological system believed to be relatively stable.

There are various causes of $^{234}U$ excess in fluid-mineral systems, which have been studied
at length (e.g. Osmond & Cowart, 1992, 2000; Porcelli & Swarzenski, 2003; Suksi et al.,
2006). In summary, $^{234}U$ is generated from α decay of $^{238}U$, and may preferentially be
increased in the fluid state during mineral-fluid interaction due to oxidation state and
valence differences between the U species (e.g. Suksi et al., 2006). Uranium activity ratios
record information on the redox state of fluids, the source of uranium in the fluids, and
potentially the timing of uranium residence in the fluid; therefore, they have long been a
focus of groundwater studies (e.g. Osmond et al., 1968; Osmond & Cowart, 2000; Porcelli
& Swarzenski, 2003). Of general interest here, is whether carbonates precipitated from
different geological settings are likely to have significant $^{234}U$ excess such that any
measured $^{238}U/^{206}Pb$ dates will be inaccurate.

Cave drip-water that generates speleothem deposits typically has excess $^{234}U$ relative to
secular equilibrium, although sometimes $^{234}U$ is depleted. Overall, most cave systems have
initial activity ratios that are not grossly offset from secular equilibrium. This means that an



uncertainty limit can be placed on such carbonates with reasonable confidence.
Disequilbrium corrections will significantly affect age estimates with high precision, but not
the low precision estimates that typically characterise LA-ICP-MS dates. For example,
Woodhead et al. (2019) used an estimate of $1.0 \pm 0.3$ for $^{234}U/^{238}U_0$ in their study of
speleothems from the Nullarbor plain, Australia, and this had negligible impact on the
resultant compilation of U-Pb dates. Hopley et al. (2019) estimated a range of $^{234}U/^{238}U_0 =$
1.26 to 2.99 for the 'Cradle of Humankind' in South Africa, with a mean of 1.9, and
discussed a resulting potential age range of 5.8 to 4.8 Ma. A known excursion from 'typical'
activity ratios is the Transvaal Dolomite Aquifer, also in South Africa. Speleothem deposits
in cave systems that interacted with water from this aquifer have anomalously high U
activity ratios ranging from ca. 2 to 12 (Kronfeld et al., 1994). This well-known occurrence
highlights that speleothem deposits could arise from fluids with variable and anomalous
activity ratios, and thus that attention must be given to accurately estimating the $^{234}U/^{238}U_0$
when dating such deposits.

Unfortunately, activity ratio data that is relevant to hydrothermal and other vein-filling
carbonates is sparse and potentially more variable. Carbonates precipitated in the shallow
crust may arise from percolating groundwater, seawater, deep brines, formation waters, or
a mixture of these sources. We can use existing data on these fluid sources to make an
initial estimate of what range may exist in terrestrial carbonates. Groundwater is well known
to have highly variable and significant $^{234}U$ excess (e.g. Osmond and Cowart, 1976). Figure
16 shows a compilation of $^{234}U/^{238}U$ activity ratios taken from a range of literature sources
(see supplementary file for sources). The population of data for groundwater (Figure 16A),
mostly shallow, but including some saline and deeper samples, has a median activity ratio
of 2.25, and is skewed towards higher values, with a significant tail up to ~11. Data from
hydrothermal fluids and deep brines are less abundant in the literature, but can be
estimated from young carbonates precipitated in travertines and hydrothermal veins. The
compilation shown in Figure 16B is skewed towards samples from Turkey and surrounding
regions.  It has a median of 1.41, and is right-skewed with a tail ranging up to ~8 and only a
few higher values.




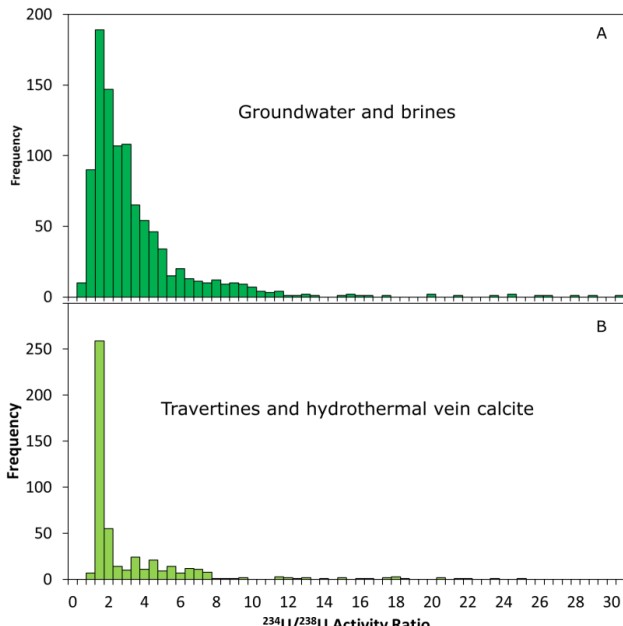



*Figure 16. Compilation of uranium $^{234}U/^{238}U$ activity ratios from the literature of: A)*

*groundwater and deep brines - these are present-day $^{234}U/^{238}U$ values (note the compilation*

*is dominated by shallow groundwater rather than brines); and B) travertines and calcite*

*precipitated in veins, commonly but not exclusively associated with travertines – these are*

*estimated $^{234}U/^{238}U_0$ values.*



The compilations in Figure 16 are somewhat alarming, as they suggest that activity ratios
have a high likelihood of being secular equilibrium ($^{234}U/^{238}U$ of ~1) in vein-filling
carbonates. The compilations shown are biased by sampling, so uncertainties on the range
of activity ratios should not be based on these compilations. However, a very conservative
view would be that shallow groundwater $^{234}U/^{238}U$ activity ratios average closer to ~2 than
they do to ~1; hydrothermal waters average closer to ~1.5; and permissible values may be
extremely out of secular equilibrium at >10. The data reveal that precise age estimates of
young carbonates derived from crustal fluids are going to be severely hampered by a lack
of knowledge of the U activity ratios.

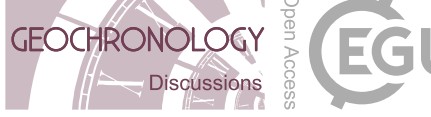

To demonstrate the effect of initial activity ratios out of secular equilibrium, we have
modelled synthetic data in Figure 17. This figure shows curves representing samples of ten
different ages, which would range from 500 ka to 9 Ma if $^{234}U/^{238}U_0$ was in secular
equilibrium (~1) during formation. The true age of the samples get younger as $^{234}U/^{238}U_0$
increases. The effect does not decrease in significance as we look at older ages, i.e. the
age offset on a sample with a measured age of 8 Ma is similar to that on a sample of 4 Ma.
The curves are shown on a log scale, because in many systems, the variation in activity
ratio is going to vary a small amount, close to secular equilibrium (~1). For example, in the
Nullarbor plain cave systems, the variation is likely to be within 30% of 1 (Woodhead et al.,
2019). Systems with large variations in initial activity ratios, for example some hydrothermal
systems, would lead to a large uncertainty on the obtained dates. Ignoring the effect of the
likely $^{234}U$ excess in vein-filling carbonates is likely to lead to significant inaccuracy of dates
by 10s of %, in general by overestimating the age. Considering the impact that
unconstrained initial $^{234}U/^{238}U$ ratios have on young dates leads to significant (> 10%)
uncertainties.

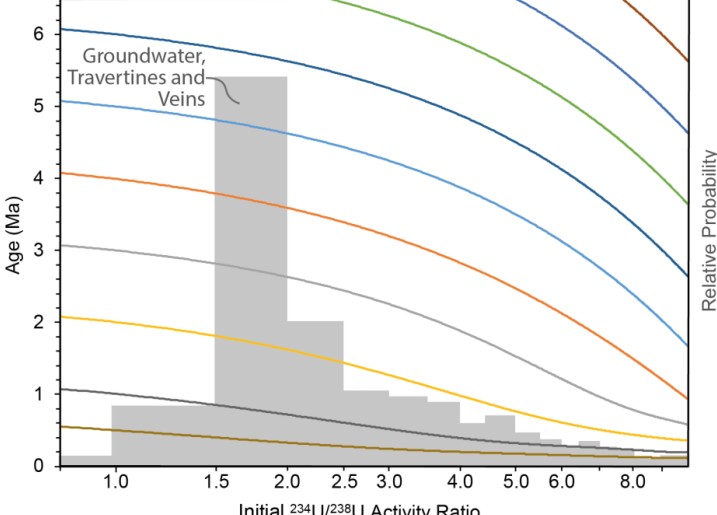

*Figure 17. Curves in different colours showing how an individual $^{206}Pb/^{238}U$ age (y-axis) will*
*vary with a change in the initial $^{234}U/^{238}U$ activity ratio (x-axis). For example, a sample*
*providing a measured $^{206}Pb/^{238}U$ age of 5 Ma will actually have a true age of 3.1 Ma if the*
*initial $^{234}U/^{238}U$ is as high as 6. The grey histogram shows the combined compilations of*
*groundwater, travertine and vein data from Figure 16.*







So far, the discussion has involved the uncertainties surrounding excess/deficient $^{234}$U
during calcite growth. However, there are several other intermediate daughter products in
the uranium decay chains that can pose problems for the accuracy of measured ages; see
Richards et al. (1998) and Woodhead et al. (2006) for previous discussion of these. The
isotope $^{230}$Th is a potential consideration in the accuracy of $^{238}$U-$^{206}$Pb ages. In general,
most speleothem-dating studies assume no initial $^{230}$Th in the system, as Th is very
insoluble in water compared to U. Any excess initial $^{230}$Th during formation would also result
in artificially old measured ages. $^{231}$Pa is another daughter product in the decay chain,
which again, is considered very insoluble, and does not form part of the disequilibrium
corrections at present. $^{226}$Ra, another intermediate product, may co-precipitate with U, but
its short half-life of 1.6 ka means it is likely to have little impact on U-Pb ages (Richards et
al., 1998). A final concern is the gas $^{222}$Rn, as this may be lost from the system by diffusive
processes. A study into the effect of this showed negligible impact on the $^{238}$U-$^{206}$Pb ages of
a Quaternary speleothem (Richards et al., 1998).

Although the effects of disequilibrium in these shorter-lived intermediate daughter products
is considered to be minor, and likely within the uncertainty of measured LA-ICP-MS U-Pb
dates, it is worth noting that hydrological systems outside of those concerning speleothems
and meteoric water have not been explored. Most of the issues presented here, particularly
the excess $^{234}$U problem, are part of the $^{238}$U decay chain, and thus have implications for
$^{238}$U/$^{206}$Pb and lower intercept ages. The $^{235}$U decay chain has different intermediate
daughter products, and thus measured $^{235}$U/$^{207}$Pb and lower intercept ages will be affected
by a different set of processes. The problem of excess $^{234}$U is alleviated if $^{235}$U-$^{207}$Pb ages
can be used instead of $^{238}$U-$^{206}$Pb ages. However, there have been few attempts to utilise
$^{235}$U-$^{207}$Pb dates (e.g. Hopley et al., 2019) because the low abundances of these isotopes in
comparison to $^{238}$U and $^{206}$Pb are major limitations on the uncertainty of the measurements.
Engel et al. (2019) have provided a solution that will potentially increase the accuracy of
age estimates for speleothems, utilising the $^{235}$U decay chain, as well as using $^{208}$Pb in
place of $^{204}$Pb as the initial lead composition. This approach is based on ID, and it is unclear
how effective it will be for LA-ICP-MS dating, given that $^{204}$Pb is difficult to measure at high
precision.



In summary, initial disequilibrium is clearly a major issue for the accuracy of U-Pb dating of
carbonates. The effect is significant for material of any age, but as we get to older
carbonates, the analytical uncertainty contributions will begin to swamp the uncertainties
surrounding disequilibrium. For dating of Neogene-Quaternary carbonates, prior knowledge
of likely activity ratios (e.g. by measuring younger or present-day values of the precipitating
fluid, and inferring no change back in time) is critical for precise and accurate dates. The
variation in hydrothermal systems that mix meteoric water with older brines is likely to be
large in terms of the degree of $^{234}$U excess. More information is needed to further
understand what sort of values can be expected in different systems and different settings.
From our preliminary compilation, it is apparent that $^{234}$U excess is the norm, rather than the
exception. For now, the absolute values and uncertainties on young dates (late Neogene to
Quaternary) with no estimation of the initial disequilibria should be treated with caution.

**10. Dating old material – dealing with a potentially open system**
Many early carbonate dating studies were attempted on very old material, i.e. Proterozoic
and Archaean (e.g. Moorbath et al., 1987; Jahn, 1998; Taylor and Kalsbeek, 1990;
Whitehouse and Russell, 1997); these mostly utilised Pb-Pb dating. A major issue of the
Pb-Pb method, is that Pb contents of crustal fluids are much higher than that of the primary
carbonates, and therefore, even small amounts of fluid-related alteration can dominate the
measured Pb-Pb composition and lead to an age that is not representative of primary
carbonate precipitation (e.g. Sumner & Bowring, 1996). Although there have been a handful
of studies dating old carbonate material since the 1990s (e.g. Ray et al., 2003; Sarangi et
al., 2004; Babinski et al., 2007; Fairey et al., 2013), Pb-Pb and U-Pb dating of Precambrian
material have become rarely used techniques. This is presumably due to the difficulty in
obtaining meaningful primary ages of old material. The dominant reason for this difficulty
can generally be distilled down to open-system behaviour, i.e. dating material that has
remained a closed isotopic system since its formation is increasingly difficult with
increasingly older material. This is simply because thermal- and/or fluid-induced mobility of
parent and daughter isotopes becomes increasingly likely if the material has been exposed
to multiple deformation-, burial-, uplift-, glaciation-, weathering- or fracture-related events.

Early studies documented various transformative processes and their impact on Pb-Pb/U-
Pb isotope systematics, e.g. fluid infiltration in limestone (Smith et al., 1991), diagenetic



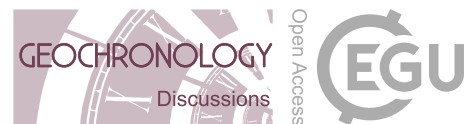

change from aragonite to calcite (Jones et al., 1995), and resetting of Pb isotope signatures
during metamorphism (Russell et al., 1996; Whitehouse and Russell, 1997; Babinski et al.,
1999). In general, the existence of some form of open-system behaviour within a given
dataset has only been recognised through the isotopic data themselves, not through an
independent dataset. This is simply achieved by assessing the robustness of the Pb-Pb or
U-Pb data array with mathematical means, e.g. using the MSWD value, and explaining
analytical scatter outside of a robust array as due to open system behaviour. With *in situ*
methods, the approaches that we have described in Section 5 may allow for some
independent removal of data that pertains to open-system behaviour, leaving a dataset that
corresponds to a closed system.

A method that has been utilised to screen for altered samples in whole-rock geochemistry,
is to test for effects of modern weathering using $^{234}U/^{238}U$ ratios (Albut et al., 2019). Ancient
samples should have measured $^{234}U/^{238}U$ activity ratios in in secular equilibrium, and
departure from this in a measured sample would imply a more recent addition or subtraction
of $^{234}U$ through weathering processes, indicating some modern fluid-rock interaction. This
method of sample screening has not been applied to U-Pb dating, but we suggest is worthy
of investigation.

In Figure 5 we documented various U-Pb datasets to demonstrate the range of behaviour
that is seen with natural carbonates. Here we provide some additional comments regarding
open-system behaviour, first in terms of U mobility, followed by that of Pb mobility. Uranium
is mobile in oxidising fluids, so U enrichment and depletion relative to Pb is assumed to be
the most common cause of open-system behaviour that will occur in natural carbonates. In
Tera-Wasserburg space ($^{238}U/^{206}Pb$ vs. $^{207}Pb/^{206}Pb$), U mobility will be apparent as sub-
horizontal trends in the data, with movement to the right reflecting gain of $^{238}U$, and
movement to the left reflecting loss of $^{238}U$ (see Figure 18). During a period of mobility,
uranium may move into a fluid-phase, such that the remaining carbonate solid remains
variably depleted in $^{238}U$, or, uranium may partially move from its original location to another
within the measured sample volume. In the former, this can sometimes be detected from
the isotopic data if a distinct departure from a robust regression is defined by a sub-
horizontal array (see Figures 5d and 18). In the latter case of uranium mobility, some
domains will be depleted, whereas others will be enriched. This may be difficult to ascertain
from the isotopic data alone if the mobility is pervasive through the material, because the



induced scatter in the U-Pb regression (from both positive and negative movement in
$^{238}$U/$^{206}$Pb) cannot be resolved from other causes of scatter, such as mixing between
different age domains.

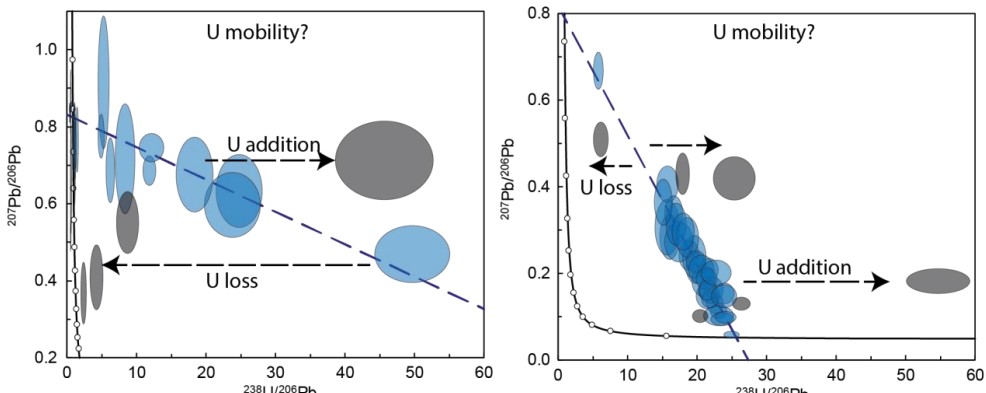


*Figure 18. Tera-Wasserburg plots for LA-ICP-MS U-Pb data from two slicken-fibre calcite*
*samples that exhibit potential open system behaviour caused by U mobility. Vectors for U*
*loss and gain are schematic. Evidence for such U mobilisation requires additional lines of*
*evidence that are currently lacking.*

Lead can substitute for Ca in the calcite lattice, and is also insoluble in most upper crustal
fluids, for these reasons, U mobility is generally considered in favour of Pb mobility. Fluid-
assisted mobility of U is certainly the most likely cause of open system behaviour because
of the solubility of some U species. However, at high temperatures, solid-state diffusion is
also a factor of consideration. Based on experimental data, Pb diffusion in calcite is
essentially slow enough to be non-existent below 300°C (when considering the composition
of a grain 1 mm in diameter; Cherniak, 1997); however, at higher temperatures (>400°C),
diffusion of lead is possible if encountered for long periods (> 20 Myrs). Empirical
observations of Pb (or U) diffusion in calcite are lacking. Diffusion is unlikely in the low
temperature calcites that have formed the basis of most modern LA-ICP-MS dating studies;
however, carbonates form in a range of higher temperature environments as well, such as
alteration veins within deeply subducted crust. Understanding how the calcite U-Pb system
works at medium to high-metamorphic grades may therefore become very relevant
information, allowing this chronometer to be used to understand dates and rates in deep
crustal environments.



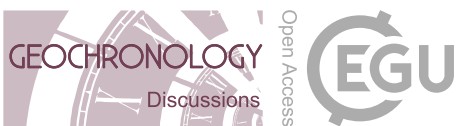



## 11. Discussion

LA-ICP-MS U-Pb carbonate geochronology has been demonstrated by this and previous
studies, to offer a potentially robust technique to date the timing of carbonate mineral
formation. Limitations on the technique arise from several challenges. These include the
typically low U content of carbonates in many settings, the propensity for carbonate to
include significant concentrations of Pb upon formation, and the ease with which fluids can
alter or reprecipitate mineral growth. LA-ICP-MS is an *in situ* technique, with high spatial
resolution compared to physical sampling for bulk dissolution studies, which enables many
of the hurdles in carbonate geochronology to be overcome. Rare and localised high U
domains can be located and sampled, a range of U/Pb ratios can be targeted to generate a
spread in isochron regressions, and altered domains, inclusions and secondary
mineralisation can often be avoided.

Accurate and informative U-Pb carbonate geochronology demands careful imaging and
petrographic analysis to establish a link between date and process. Various imaging
techniques can be utilised prior to or after dating to aid with mineral characterisation, and
with refinement and interpretation of the resulting age data. We refer to this as image-
guided analysis. An alternative technique involves directly determining age data from
image-based data itself, which we refer to as image-based analysis. Both techniques have
their different benefits and applicability, and their efficacy depends on the instrumentation
used and the type of material; for example, quadrupole ICP-MS is suited to image-based
analysis, as a large element suite can be measured. Limitations on using quadrupole
instrumentation are the detection limits for U and Pb when counting a large suite of
elements. In contrast, multi-collector instruments can be used for image-based analysis,
and have a very low detection limit, but the mass range is restricted between Hg and U,
meaning that additional elements useful for understanding the U and Pb distribution cannot
be measured simultaneously. Overall, image-based analysis is only nascent in
geochronology, and as such has not been fully explored.



### 11.1. Limitations

There are several limitations to LA-ICP-MS U-Pb carbonate geochronology. The heterogeneous nature of carbonate materials pose a problem that is difficult to circumvent. The relatively high spatial resolution of laser ablation already offers the best solution to this problem, but detection limits and the very low U and Pb contents mean that spots >150 μm are commonly employed, hampering the full ability of laser ablation to target fine-scale (< 50 μm) zonation. Improvements in efficiency of ICP-MS instrumentation and of the ablation process are possible solutions to his issue.

At present, there is only one reference material in circulation that has been widely used and documented for the purpose of U-Pb normalisation (WC-1; Roberts et al., 2017). WC-1 has an uncertainty on its U/Pb ratio of 2.5% $2\sigma$. Using this material for normalisation of U/Pb ratios, or for validation of the method accuracy, limits the final age uncertainty of any particular sample to ~2.5%. To improve beyond this range requires the characterisation of natural (or production and characterisation of U and Pb doped synthetic) materials, with a final U/Pb precision better than 2.5%. There is also a requirement for additional well characterised materials (i.e. those with robust U-Pb systematics and well documented ID U-Pb datasets) that can be used as secondary reference materials (i.e. those run as unknowns), for assessment of accuracy and long-term reproducibility.

Another major limitation is the nature of carbonate matrices, and the lack of quantified data on the matrix effect between different carbonate minerals and structures. Inter-element fractionation (i.e. U/Pb in this case) is one of the major limitations on the reproducibility and accuracy of laser ablation U-Pb dating. For this reason, matching matrices of the reference material with that of the sample has been standard practise in U-bearing accessory mineral geochronology. Several groups have tried to limit the effect of this issue by utilising normalisation and date reduction procedures that reduce the effect (e.g. Burn et al., 2017; Neymark et al., 2018), but regardless of the matrix used for normalisation, validation of the method should still utilise a similar matrix to the sample. Carbonates clearly have a large range of structures, even with calcite, for example, sparry to micritic, with wide-ranging crystal/grain-sizes and porosity. Nuriel et al. (2019) noted differences between the use of coarse-grained sparry reference materials to fine-grained polycrystalline reference materials, with the latter being skewed towards older ages by several percent. To move





towards better precision and accuracy of the LA-ICP-MS U-Pb method, it will be necessary
to have a range of well characterised reference materials that cover variable carbonate
mineralogy (e.g. aragonite, dolomite, calcite), as well as internal morphology and texture.

11.2.    Applications
To date, LA-ICP-MS U-Pb carbonate geochronology has been applied to a wide range of
applications. These include the dating of speleothem deposition (Hopley et al., 2019;
Scardia et al., 2019), brittle deformation (Roberts & Walker, 2016; Ring & Gerdes, 2016;
Goodfellow et al., 2017; Hansman et al., 2018; Parrish et al., 2018; Beaudoin et al., 2018;
Nuriel et al., 2017, 2019; Smeraglia et al., 2019), hydrocarbon migration (Holdsworth et al.,
2019), hydrothermal ore mineralisation (Burisch et al., 2017, 2019), hydrothermal fluid flow
(Mazurek et al., 2018; Walter et al., 2018; Incerpi et al., 2019; MacDonald et al., 2019);
pedogenesis (Methner et al., 2016; Liivamägi et al., 2019); ocean crust alteration (Coogan
et al., 2016), diagenesis in sedimentary deposits (Li et al., 2014; Pagel et al., 2018;
Mangenot et al., 2018; Godeau et al., 2018; Lawson et al., 2018) and sedimentary
deposition (Drost et al., 2018). Published dates range in age from 0.6 to 548 Ma (see
Figure 19), MSWDs range from 0.2 to 89 (Figure 19a), and quoted uncertainties range from
0.6 to 143 % (2s; Figure 19B). The majority of dated samples so far range from the
Neogene to Jurassic, with ~50% being Oligocene or younger. Across this age range, the
uncertainty is variable and uncorrelated to age or MSWD, demonstrating that the age
uncertainty reflects an interplay of factors, and includes the heterogeneous nature of
carbonate materials. It should be noted however, that many dates with large uncertainties
or mixed results are likely unpublished, biasing this compilation towards successful
samples. For example, it is possible that many unreported and failed attempts at dating
samples that are Palaeozoic and older have been made.






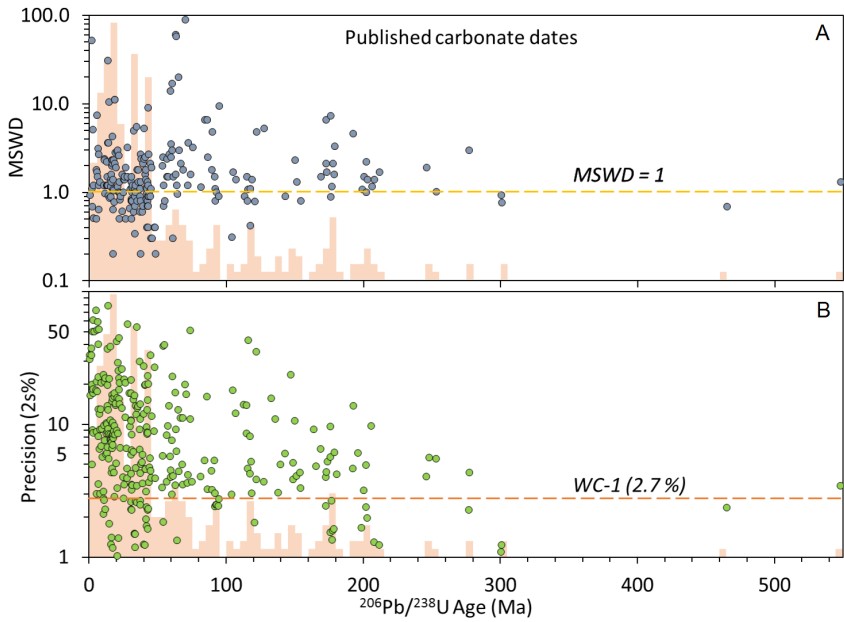

*Figure 19. Compilation of published LA-ICP-MS U-Pb dates of carbonate (n=318). A)*

*MSWD plotted against $^{206}Pb/^{238}U$ age; and B) Precision as 2s % plotted against $^{206}Pb/^{238}U$*

*age. The histograms in the background show the distribution of dates.*

A major benefit of carbonate geochronology is that carbonate minerals record an archive of

data that can be linked to the age of formation. Fluid inclusions, stable isotopes (carbon and

oxygen), radiogenic isotopes (strontium), and elemental compositions all reveal insight into

the fluid composition that precipitated the mineral. This combination has long been an

approach within the field of palaeohydrology; however, the timing of mineralisation and

hence fluid-flow has generally involved only relative estimates with large uncertainties, or

the dating of phases associated with higher-temperature activity (e.g., Re-Os dating of

Molybdenite). The addition of absolute chronological information is a critical step to

understand the timing of fluid-flow through the crust in a range of settings, for example,

within hydrocarbon-bearing basins, within ore-forming mineral systems, and within upper

crustal bedrock that may be used to host anthropogenic waste/outputs (e.g. radioactive

waste, storage and sequestration of $CO_2$).





A benefit of utilising LA-ICP-MS as a method of dating, is that the same crystals that have
been dated can be measured for various other chemical proxies and signatures. Several
previous studies have combined fluid inclusions and/or stable carbon and oxygen isotope
analysis with LA-ICP-MS dating (e.g. Mangenot et al., 2018; Pagel et al., 2018; Goodfellow
et al., 2016; Walter et al., 2018), but for most of these, it is not clear if the same volume of
material, or simply the same genetic domain has been sub-sampled for both the dating as
well the additional isotope analyses. Use of petrography and imaging allows for the same
genetic domain to be analysed for several methods; however, there are also several
approaches that allow for an overlapping analytical volume to be analysed. Dated material
can be micro-drilled or -milled following laser ablation, with the powder being analysed for
additional chemical information (e.g. Sr, C, O isotopes). Alternatively, thin sections or
polished blocks can be analysed using a combination of in situ techniques, for example, ion
microprobe measurement of stable isotope and/or elemental compositions, and laser
ablation measurement of Sr isotopes, elemental compositions along with U-Pb dating.
Drake et al. (2017) demonstrated the utility of combining ion microprobe stable carbon and
oxygen isotope analysis with U-Pb dating to study palaeohydrology and ancient microbial
activity.

In addition to traditional carbon and oxygen isotope measurements ($\delta^{13}C$ and $\delta^{18}O$),
clumped isotopes ($\Delta 47$) can provide the temperature of mineral formation (e.g. Eiler, 2007).
Several studies have demonstrated the combination of clumped isotope thermometry with
dating (e.g. Quade et al., 2018; Mangenot et al., 2018; Lawson et al., 2017; MacDonald et
al., 2019). These apply the technique to the dating of paleosols for climatic records,
diagenetic mineralisation for basin histories, and hydrothermal veins to understand crustal
fluid-flow. This combination of techniques is a clear growth area with a range of applications
across earth and environmental science.

Finally, carbonates also comprise a host of major and trace metals that offer further isotopic
information that has yet to be fully explored, for example, stable isotopes of Ca, Zn, Fe, and
Cu. Linking these with U-Pb dates from the same material could provide high resolution
records of natural fractionation processes in subsurface environments.

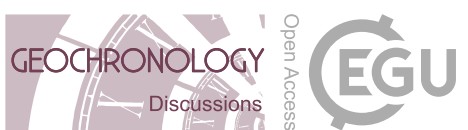

## 12. Conclusions

We have demonstrated the heterogeneous nature of carbonate minerals, in terms of U and Pb distribution and isotopic systematics. Although we have focused on vein-filling calcite, most of the issues highlighted in this paper are relevant to other carbonate dating applications. Various imaging techniques can be used to screen material, and to characterise U-Pb heterogeneity; a combination of these techniques is crucial to understand what exactly has been dated. Linking age information to spatial data, imagery or elemental maps, is crucial to understanding heterogeneous isotopic data. Two main approaches to dating have been presented, the traditional approach of static spot ablations guided by independent image data, and an alternative approach of age mapping using software analysis of 2D isotopic map data. Each of these approaches have benefits and drawbacks, and the choice between them will partly be governed by the instrumentation available. The applications of carbonate U-Pb geochronology are vast, with a key benefit to the laser ablation approach being that specific volumes of material can be analysed for several isotopic and elemental proxies and signatures, whilst also providing absolute chronological information. The LA-ICP-MS method is limited by factors that include the uncertainties on reference material isotope ratios, matrix effects and long-term reproducibility; taking these into consideration, the method is best applied to applications where age uncertainties of greater than 3-4% are of benefit. For applications where high precision (i.e. <1%) is required, such as calibration of palaeoclimate records or of evolutionary change, then follow-up analysis with ID is the only method that can potentially achieve the necessary precision. The future of the method in terms of accuracy and precision requires well characterised (by Isotope Dilution methods) reference materials covering a range of carbonate matrices. The range of studies published over the last five years (2014 to 2019) have revealed a wide array of geoscience applications that are both amenable to, and benefit from, LA-ICP-MS U-Pb carbonate geochronology.

## 13. Acknowledgements

The authors acknowledge the Natural Environment Research Council for National Capability funding of the National Environment Isotope Facility. N Roberts, D Condon, M Horstwood, A Milodowski and R Haslam publish with the permission of the Executive Director of the British Geological Survey. D Chew and K Drost acknowledge a Science Foundation Ireland grant (15/IA/3024) that is partly funded by the Geological Survey of




Ireland and the Environmental Protection Agency, Ireland. HD acknowledges Swedish
Research Council grant (2017-05186) and Formas grant (2017-00766).
**14. Appendix**
14.1.      Implications of age data
The focus of this paper is not on the meaning of the age data presented, or its implications
for faulting or fluid-flow; however, we provide brief information for interested readers.
14.2.      Example A and D - Variscan-related veins in the Northumberland Basin
The age of ca. 287 Ma for the dated calcite crystal can be linked to deformation of the host
rock based on the vein structure. The calcite is taken from a planar fracture forming on the
axial plane of a small fold that has accommodated bedding-plane sliding (Fig. 8). The
fracture is filled with calcite mineralisation of the stretched vein type (Bons et al., 2012), and
that is interpreted to have formed soon after opening of the vein, and synchronous with
deformation. The age of ca. 287 Ma broadly overlaps with the intrusion of the Whin Sill (ca.
297 Ma; Heaman pers. comm. within De Paola et al., 2005), and is therefore compatible
with the model of partitioned transpression of De Paola et al. (2005), who suggest that
deformation was synchronous with the Whin Sill intrusion.
14.3.      Example B - Faroe Island brittle faults
The significant of the Eocene ages has been discussed by Roberts & Walker (2016). This
paper was the first to demonstrate the applicability of LA-ICP-MS U-Pb carbonate
geochronology to dating brittle structures in the upper crust.
14.4.      Example C - Sellafield fracture mineralisation
Sample 877 was collected from the modern-day saline transition zone between the upper
fresh groundwater system and the deeper saline basinal-basement groundwater system, at
a depth of -635 m OD within the St Bees Sandstone Group (Triassic) in Sellafield borehole
BH10A (equivalent to sample B697 and D750: Appendix Table S2, Milodowski et al., 2018).
Externally, this calcite exhibits a "nailhead" (i.e. c-axis flattened) crystal habit (Figure 10).
However, detailed petrographic analysis reveals it has a complex growth history:



comprising of cores of c-axis-elongated calcite characteristic of the deeper saline
groundwater that are syntaxially-overgrown by later equant and c-axis flattened calcite
characteristic of the overlying fresh groundwater zone (Milodowski et al., 2018). The U-Pb
analyses all come from within the saline groundwater zone type calcite core region (rather
than the later freshwater-type overgrowth that has extremely low U).

Late-stage (generation "ME9") calcite is a characteristic feature of the present-day fracture-
controlled deep groundwater system in the Sellafield area of the west Cumbrian coastal
plain (Milodowski et al., 2018). The resulting age suggests that ME9 calcite growth in the
sampled fracture was initiated in the late Miocene, and has been preserved (or at least
partially preserved until the present-day). The implication is that the modern groundwater
system was developed following regional Miocene uplift and younger groundwater recharge
relating to glaciations and/or uplift of the region, have not led to complete re-precipitation of
fracture-filling calcite, with calcite precipitation continuing to the present-day. Taken
together with other petrographic, stable isotope, strontium isotope, fluid inclusion,
microchemical analyses and whole-crystal U-Th age dating, the age data support the
interpretation that despite evidence for glacial recharge, the geochemical conditions (e.g.
pH, Eh) have remained stable over this period at potential repository depths (cf. Milodowski
et al., 2018).

14.5.    Example E and F – Vein sets of the Bighorn Basin, Wyoming
These samples are from vein sets in the sedimentary cover of the Bighorn Basin. These
samples are part of a larger study that analysed the timing of deformation in the foreland of
the Sevier and Laramide orogenies, and how this deformation propagated in time and
space (Beaudoin et al., 2018). Sample BH11 is related to Laramide deformation, whereas
sample BM18 is related to Sevier deformation.

14.6.    Example G - Moab fault
This sample comprises multiple thin (1 to 5 mm wide) veins collected from the footwall
damage zone of the Moab Fault in southeast Utah. Regional deformation is primarily driven
by salt tectonics (Gutierrez, 2004), and salt dissolution has produced up to one km of offset
within the sedimentary rocks along the Moab Fault (Foxford et al., 1996). Fault zone
deformation was closely associated with fluid flow and carbonate cementation (Eichhubl et



al., 2009; Hodson et al., 2016). Ar-Ar ages from clay fault gauge range from 63 to 43 Ma
and are interpreted to record the final episodes of faulting and fracture generation (Pevear
et al., 1997; Solum et al., 2005). Our new lower intercept age of 22 Ma is imprecise, but
clearly younger than the early-Tertiary ages. This suggests that circulating fluids continued
to move along the fault zone long after the cessation of fault related deformation.

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
