# Peer review of "LA-ICP-MS U-Pb carbonate geochronology: strategies, progress, and"

_Geochronology, 2019_

## Referee Comment (RC1) · Andrew R Kylander-Clark (Referee) · 12 Dec 2019

"LA-ICP-MS U-Pb carbonate geochronology: strategies, progress, and application to fracture-fill calcite" is a well-organized, well-written manuscript that describes, in considerable detail, current methodologies and applications in dating calcite by laser ablation. It is likely to be a long-lived reference paper for anyone interested in dating calcite, as it contains many relevant examples, and an exhaustive list of our current understanding of the many aspects of calcite geochronology. I have a few minor comments that I believe could improve the original manuscript, but feel that the authors have already done a thorough job producing this work. The comments, related to line

numbers, are as follows:

122: Isn't point 2 the same as point 1? You need more sample to get higher sample/blank ratios.

259: Is this really true? For example, do we know the absolute age constraints of WC1 better than Ash15 or Duff Brown? They are younger, but a 5% uncertainty on Ash15 is only 150kyr. Once counting statistics get better than a few percent, the increased precision is moot. It is true that secular equilibrium uncertainties can punish younger ages more in a relative sense; you may want to point to that part of the discussion here.

274: This is the main point here, which should be highlighted. When the data is closer to concordia, there is less variability in the intercept age no matter what the common component is. That is, your assumption on a fixed common value is less important when the samples are older. Nevertheless, if you can assume a fixed common component, or you have a large spread in $\mu$, I'm not sure your confidence is better with an older sample (in an absolute age sense).

What might be nice in the figure is to show the relative and absolute age uncertainties on each isochron given either a fixed U concentration or fixed common Pb concentration. You could also use a fixed analytical percentage, but the younger samples might have worse analytical precision due to poor counts. Nevertheless, this addition would be more elucidating.

296: Why do you say inaccurate here instead of imprecise?

Figure 6: Here it would be nice to show a median value for other U- and Pb-bearing geochronometers as a comparison. I realize this is a tough ask, but you could take say apatite and or titanite from a paper that studied a range of samples.

644: I presume this is after repolishing?

1062: This sentence is awkward.

1067: delete "they do"
* * *
**GChronD**

---

## Referee Comment (RC2) · Troy Rasbury (Referee) · 18 Dec 2019

Review of Roberts et al. This manuscript is sort of a funny mix of a review and new datasets. While the title suggests it is about dating veins, much of the text is devoted to other types of carbonate dating. While it is nominally a review of vein dating, it is not a complete review of what is out there. My comments below are really specific to the sections I am making the comments on, but my overall take on the manuscript is that it could be more compactly packaged and could do a better job of reviewing all of the U/Pb dating of vein material- perhaps leaving off discussion of speleothems which is well covered by Woodhead and a laundry list of carbonate characterization techniques

that aren't tied to examples. The modeling of initial disequilibrium is a very important contribution. Why is it so late in the paper? I think it should be upfront and that the ages that are being reported from various veins might be considered in light of this. Does the few million years matter? What is the question- and how accurate does an age need to be to answer it.

Introduction- It is clear that laser ablation dating of carbonates is going to take U-Pb carbonate dating to a new level. It is particularly clear that using the Drost et al. approach for extracting pixels from imaging is going to push the potential even further, and in terms of uncertainties, perhaps to the realm where it competes with isotope dilution. I feel like the either or aspect to this introduction is a bit off track. That is, it is clear one can get far more precise measurements with isotope dilution. The size of the error ellipses at the best one can do with laser ablation is far larger than for isotope dilution. Obviously, the system has to be pretty well behaved to give a good line and the uncertainties are a reflection of that. However, the error ellipses are so large that the MSWD should be very low and even so there is often over dispersion of data. So even though one can drive the uncertainties down by increasing the data spread in a very clever way, I do not believe that this in anyway replaces the importance of isotope dilution for establishing the reliability of these ages. This is addressed further into the document, but it is a little bit overstated in the Introduction in my opinion. Figure 1: This should include a concentration scale. There should also be plane light and crossed polar light photomicrographs of the sample to go with the laser ablation maps. Further, it would be useful to see these maps with respect to other elements, particularly Fe, Mn, Sr, and Mg- the usual calcite elements measured for comparison. This will allow researchers who are interested in dating their carbonates to compare to what you have successfully dated.

Figure 2: This is a nice figure. I think a stronger point could be made that based on the LA results it would be possible to sample the high and low mu areas and achieve closer to the same spread in the data with ID. Presumably, you have synthetic data that could

be used to do more with this figure. For example is Figure 2B the U concentration? The U/Pb? What is the concentration? From the description you hold Pb constant and have a range of U. What did you use for the initial Pb? Does it have scatter? How did you create the data used for the two isochrons? I am wondering why there is so much scatter in the green points for the LA data?

Line 300-Just a personal preference, but I think starting a sentence with a call out to a figure is awkward. Instead, I recommend making a statement about what the figure indicates and then calling the figure out. I won't repeat this for the remaining examples, but I think this should be changed throughout the text. I would also break this up into multiple sections which could call out figure 5 rather than as this lab project looking list you have. Line 358 is not accurate. There is no indication that having reduced U is more stable or more common than oxidized U. Both oxidized and reduced U have been observed and the U/Pb systematics, and even the U concentrations have not been shown to be controlled by this. Line 366 is also not accurate. U(IV) is insoluble in reducing fluids but is easily oxidized when oxidizing fluids interact with it. The Drake et al 2018 result on high concentrations of U assuming that the U is reduced and that explains elevated concentrations is not supported. It could be that the U is reduced, but stating that here without real evidence is propagating a myth. A really important question is how U(IV) could remain in solution to be incorporated in carbonates? The is particularly important for vein calcites as fluids can be meteoric, or have a deep source, or may be a mix of these. I bet you can't find a speleothem calcite that doesn't have U(VI) and yet your figure and discussion shows that they have equally high to higher U concentrations to veins. Line 411 is not exactly correct. Even if a fluid goes through a Pb rich rock it doesn't necessarily become high in Pb. It depends on the solubility. An example of one of the highest mu's ever measured is from David Richards, 1998 paper from speleothems in Winnats Head Cave, Peak District UK which I understand is near a galena mine.

If the discussion of the various techniques for detailed investigation of carbonate textures and element zoning could be put into the context of their application to dating veins rather than leaving that to the imagination (I am thinking back to figure 2 here) I think it would be a whole lot more digestible for researchers who are wondering how to get started. If you don't think phosphor imaging is good for this application, you could state that in a sentence rather than discussing how it works. This has been published and is used by others. The statement that speleothems have more U than other samples is not supported by your own figure 6.

Line 519- The title destructive techniques sounds so final! A small line left by a laser ablation traverse is hardly visible on a thin section or slab and as the paragraph accurately states can give important information on the potential for U/Pb dating. Line 605- it could result in meaningless ages- but this isn't required Line 870- I would be a little careful about the wording. Pb derived from ancient source will have a high 207/206. I think you mean instead source that have a relatively high U/Pb with a long enough evolution to give lower 207/206 which is then leached or dissolved and incorporated as initial Pb in the calcite you are dating. In carbonates an easy way to do this is to somehow preserve aragonite through early diagenesis and then have it exposed to a fluid which will quickly dissolve it and take the radiogenic Pb that it produced (as well as the U). Line 906- the example of the Moab fault seems out of place. I don't understand why it follows a discussion of common Pb corrections. Line 1063 I think you mean have a high likely of not being in secular equilibrium. Your discussion at the bottom of page 42 makes a good case for ID for samples that you want to know more about secular disequilibrium in. Line 1289 I think you mean data instead of date Line 1303 applied to a range of applications – should be reworded Line 1329 record and archive say the same thing. Pick one.

---

## Author Comment (AC1) · 22 Jan 2020

"LA-ICP-MS U-Pb carbonate geochronology: strategies, progress, and application to fracture-fill calcite" is a well-organized, well-written manuscript that describes, in considerable detail, current methodologies and applications in dating calcite by laser ablation. It is likely to be a long-lived reference paper for anyone interested in dating calcite, as it contains many relevant examples, and an exhaustive list of our current understanding of the many aspects of calcite geochronology. I have a few minor comments that I believe could improve the original manuscript, but feel that the authors have already done a thorough job producing this work.

[Figure]

We thank the reviewer for these positive comments.

"The comments, related to line C1 numbers, are as follows: 122: Isn't point 2 the same as point 1? You need more sample to get higher sample/blank ratios."

Not exactly, the averaging effect of larger samples, is different from the effect of lowering the blank/sample ratio. We have reworded slightly to make this clearer.

"259: Is this really true? For example, do we know the absolute age constraints of WC1 better than Ash15 or Duff Brown? They are younger, but a 5% uncertainty on Ash15 is only 150kyr. Once counting statistics get better than a few percent, the increased precision is moot. It is true that secular equilibrium uncertainties can punish younger ages more in a relative sense; you may want to point to that part of the discussion here."

We're slightly unsure of the context of AKC's point here. The statements in our text are about theoretical precision in relation to the abundance of radiogenic lead compared to common lead. Our statements regarding this theory are correct – the fact that precision is limited in nature by other means is not relevant at this point.

"274: This is the main point here, which should be highlighted. When the data is closer to concordia, there is less variability in the intercept age no matter what the common component is. That is, your assumption on a fixed common value is less important when the samples are older. Nevertheless, if you can assume a fixed common component, or you have a large spread in $\mu$, I'm not sure your confidence is better with an older sample (in an absolute age sense)."

Yes we agree that the point about not relying on the common lead composition when data is near concordant is important, so have added an extra sentence to this section. We did not claim that older samples have more confidence – we stated that older samples with a fixed mu will have a greater abundance of radiogenic lead – this will indirectly provide more confidence to the age determination (if all other factors are

removed).

"What might be nice in the figure is to show the relative and absolute age uncertainties on each isochron given either a fixed U concentration or fixed common Pb concentration. You could also use a fixed analytical percentage, but the younger samples might have worse analytical precision due to poor counts. Nevertheless, this addition would be more elucidating."

The aim of this figure is simply to demonstrate the correlation between age and abundance of radiogenic lead – from speaking to a lot of different users in the community, this basic level of understanding of $\mu$ and ingrowth of Pb would benefit from simple explanation. If we add in uncertainties then this will add a layer of complexity that will take away from the message. To do this with synthetic data would in fact be rather difficult.

"296: Why do you say inaccurate here instead of imprecise?"

Good point. In general, imprecision is the problem. However, we probably hand in mind the many regressions we have seen that have slopes dominated by one or two data-points, that are likely inaccurate as well as imprecise. Since this is not quite what the statement is about, we have reworded to imprecise and even inaccurate.

"Figure 6: Here it would be nice to show a median value for other U- and Pb-bearing geochronometers as a comparison. I realize this is a tough ask, but you could take say apatite and or titanite from a paper that studied a range of samples."

We have updated the figure with 2D KDE plots, with greater data density, and added median values for apatite and zircon.

"644: I presume this is after repolishing?"

Yes, but the maps only remove a few microns, so after a light polish to remove these troughs, the mapping is still deemed to be pretty accurate representation of the material.

"1062: This sentence is awkward."

Reworded.

**Vein (Mid-Ocean Ridge)**
Median U ppm: 0.029
Median Pb ppm: 0.003
n=3566

apatite    zircon

**Vein (Terrestrial)**
Median U ppm: 0.061
Median Pb ppm: 0.023
n=15097

**Biogenic**
Median U ppm: 0.152
Median Pb ppm: 0.047
n=243

**Diagenetic**
Median U ppm: 0.406
Median Pb ppm: 0.350
n=2814

Pb ppm

**Speleothem**
Median U ppm: 1.878
Median Pb ppm: 0.003
n=2238

U ppm

high

data density

low

U ppm

**Fig. 1.** Updated Figure 6

---

## Author Comment (AC2) · 22 Jan 2020

Review of Roberts et al. "This manuscript is sort of a funny mix of a review and new datasets. While the title suggests it is about dating veins, much of the text is devoted to other types of carbonate dating. While it is nominally a review of vein dating, it is not a complete review of what is out there."

This is a correct understanding of the paper – most of our experience is from vein-filling carbonates, but is relevant to all carbonates. We do not wish to have those working in non-vein filling carbonates to ignore the paper, as we feel it has relevant info for all. Equally, we admit that not all carbonates are covered equally. However, with paper like

[Figure]

Woodhead & Petrus coming online – we feel the gaps will be covered by other papers in the special issue.

"My comments below are really specific to the sections I am making the comments on, but my overall take on the manuscript is that it could be more compactly packaged and could do a better job of reviewing all of the U/Pb dating of vein material- perhaps leaving off discussion of speleothems which is well covered by Woodhead and a laundry list of carbonate characterization techniques that aren't tied to examples."

We disagree – the paper is going to be a reasonable length even if we cut a couple of sections, so we do not feel that removing sections will benefit the paper – they will likely be of use to someone. Woodhead & Petrus covers a scattered amount of material, certainly not an exhausting review of speleothems. Our paper does not conflict in particular, so we feel that everything we cover has its utility. We agree that the carbonate imaging section is not tied to examples, but most techniques are then covered in the subsequent sections where several imaging techniques are used. Since there is no current literature reviewing these techniques we feel it is reasonable to keep this section.

"The modeling of initial disequilibrium is a very important contribution. Why is it so late in the paper? I think it should be upfront and that the ages that are being reported from various veins might be considered in light of this. Does the few million years matter? What is the question- and how accurate does an age need to be to answer it."

We agree, it is important. The current version could be ordered in several ways, but it our opinion follows a logical progression. Yes, we agree with the rhetorical question 'does the few million years matter?...'. This is why we have included statements such as the following: "In fact, after propagation of all relevant uncertainties, final U-Pb dates typically exceed 3% precision ($2\sigma$). For this reason, LA-ICP-MS carbonate U-Pb geochronology is particularly suited for applications in tectonics and crustal fluid-flow, but commonly less suited for applications in stratigraphy and palaeoclimate."

"Introduction- It is clear that laser ablation dating of carbonates is going to take UPb carbonate dating to a new level. It is particularly clear that using the Drost et al. approach for extracting pixels from imaging is going to push the potential even further, and in terms of uncertainties, perhaps to the realm where it competes with isotope dilution. I feel like the either or aspect to this introduction is a bit off track. That is, it is clear one can get far more precise measurements with isotope dilution. The size of the error ellipses at the best one can do with laser ablation is far larger than for isotope dilution. Obviously, the system has to be pretty well behaved to give a good line and the uncertainties are a reflection of that. However, the error ellipses are so large that the MSWD should be very low and even so there is often over dispersion of data. So even though one can drive the uncertainties down by increasing the data spread in a very clever way, I do not believe that this in anyway replaces the importance of isotope dilution for establishing the reliability of these ages. This is addressed further into the document, but it is a little bit overstated in the Introduction in my opinion."

We understand TR's point here. In general we agree. We have added reworded the caveat sentence, strengthening it, and have added another final sentence leaving the reader clear that ID is required for ultimate reliability. However, we add that if the community had not invested time into the LA technique, and had stuck with ID alone, we would simply not be able to date the types of material we now can.

"Figure 1: This should include a concentration scale. There should also be plane light and crossed polar light photomicrographs of the sample to go with the laser ablation maps. Further, it would be useful to see these maps with respect to other elements, particularly Fe, Mn, Sr, and Mg- the usual calcite elements measured for comparison. This will allow researchers who are interested in dating their carbonates to compare to what you have successfully dated."

We have added concentration scales. These are all mounted chips of calcite rather than thin sections – and thus optical imagery is limited in its usefulness. We have added photomicrographs of the samples to an additional supplementary figure however

– not all of the samples had high resolution imagery that was suitable for an in-paper figure. The lead author is collating another paper concerning trace elements in calcite, and such comparisons between elements and uranium in particular will be the subject of that paper. Several of the examples later in the paper show maps from several elements to allow comparison.

"Figure 2: This is a nice figure. I think a stronger point could be made that based on the LA results it would be possible to sample the high and low mu areas and achieve closer to the same spread in the data with ID. Presumably, you have synthetic data that could be used to do more with this figure. For example is Figure 2B the U concentration? The U/Pb? What is the concentration? From the description you hold Pb constant and have a range of U. What did you use for the initial Pb? Does it have scatter? How did you create the data used for the two isochrons? I am wondering why there is so much scatter in the green points for the LA data?"

The LA data is a natural dataset (with natural Pb-Pb variability). The ID dataset is generated by averaging these 'real' data, as per the map figure in a broad context (obviously the data pertaining to the spots will not exactly match what would be measured for ID). The uncertainties for the ID data are randomly applied with a mean of 0.5% for both Pb-Pb and Pb-U ratios. Yes, Figure 2b is the U concentration, or the U-Pb ratio - with a fixed Pb concentration they are one and the same. The concentration is low – meaning that the volume needed for low blank ID is accurately drawn. Why is there scatter – it is a natural sample – i.e. we do not know why! Variable initial lead?,Some open-system behaviour?, etc. This is irrelevant as it would be replicated regardless of the LA or ID approach.

"Line 300-Just a personal preference, but I think starting a sentence with a call out to a figure is awkward. Instead, I recommend making a statement about what the figure indicates and then calling the figure out. I won't repeat this for the remaining examples, but I think this should be changed throughout the text. I would also break this up into multiple sections which could call out figure 5 rather than as this lab project looking list

you have."

We disagree. Keeping text as simple as possible is the best approach to any paper (whether I, the lead author, manages that I cannot say). We feel that this comparison of different isochrons (the good, the bad and the ugly), has not been covered for any U-Pb chronometer – and is a useful standalone section. 'Pedagogic' – to quote two separate co-authors.

"Line 358 is not accurate. There is no indication that having reduced U is more stable or more common than oxidized U. Both oxidized and reduced U have been observed and the U/Pb systematics, and even the U concentrations have not been shown to be controlled by this."

Our mistake, we have corrected this sentence. (We were originally incorrectly alluding to the fact that U(IV) should be more stable due to its size – i.e. the Sturchio et al 1998 study, but we realise this is not what was said in the cited Rasbury and Cole paper.

"Line 366 is also not accurate. U(IV) is insoluble in reducing fluids but is easily oxidized when oxidizing fluids interact with it. The Drake et al 2018 result on high concentrations of U assuming that the U is reduced and that explains elevated concentrations is not supported. It could be that the U is reduced, but stating that here without real evidence is propagating a myth. A really important question is how U(IV) could remain in solution to be incorporated in carbonates? This is particularly important for vein calcites as fluids can be meteoric, or have a deep source, or may be a mix of these. I bet you can't find a speleothem calcite that doesn't have U(VI) and yet your figure and discussion shows that they have equally high to higher U concentrations to veins."

Regarding the Drake study – firstly, our sentence importantly contains the term 'interpreted as'. We do not feel this is perpetuating a myth. Secondly, we agree that U(IV) has no current evidence - however, ongoing work is aiming to prove that. We added the term 'in these environmental conditions' – to the sentence about partition coefficients, as a caveat. We do not imply that U(IV) stays in solution in the text – this is

the reviewer's extrapolation. High U zones in calcite may in fact have acted like reducing fronts on a very small-scale – i.e. when U is reduced, it may then be in solution for a very short time – precipitating into calcite. The statement about speleothems is not pertinent – as we make no great claims about U(VI) versus U(IV), we merely state that studies have found both. In fact, we finish this section stating that: "It is evident that more data from natural carbonates in different settings are needed to more fully understand the controls on U and Pb incorporation."

"Line 411 is not exactly correct. Even if a fluid goes through a Pb rich rock it doesn't necessarily become high in Pb. It depends on the solubility. An example of one of the highest mu's ever measured is from David Richards, 1998 paper from speleothems in Winnats Head Cave, Peak District UK which I understand is near a galena mine."

Fair point. We based this on limited experience – but we point out that all hydrothermal vein samples we have dated from nearby or within Pb-bearing ore deposits were undateable due to high Pb/U ratios. Speleothems and tufas form in a different way to hydrothermal veins, and thus perhaps behave differently in terms of their mineral-fluid U/Pb behaviour. We now state that enrichment in Pb/Ca ratio may potentially take place.

"If the discussion of the various techniques for detailed investigation of carbonate textures and element zoning could be put into the context of their application to dating veins rather than leaving that to the imagination (I am thinking back to figure 2 here) I think it would be a whole lot more digestible for researchers who are wondering how to get started. If you don't think phosphor imaging is good for this application, you could state that in a sentence rather than discussing how it works. This has been published and is used by others. The statement that speleothems have more U than other samples is not supported by your own figure 6."

The examples in the paper in the subsequent sections show the combination of several imaging methods with data collection and interpretation. For some of the image

techniques that are available and that we describe, we do not have specific examples of where we found them useful. However, that does not mean other users may not find them useful themselves. For this reason, we feel it is appropriate to document all of the standard, traditional and accessible techniques that are available to the community. This technique is still growing, and thus other or better uses of such techniques may be found in the future. Of course there are other techniques such as synchrotron based, but these are accessible to very few users without specific grant projects and needs. The median U content in speleothems is 1.8 ppm, and in other samples, namely veins is 0.03 and 0.08 ppm – so yes the figure does support this statement.

"Line 519- The title destructive techniques sounds so final! A small line left by a laser ablation traverse is hardly visible on a thin section or slab and as the paragraph accurately states can give important information on the potential for U/Pb dating."

Destructive and non-destructive are commonly used terms by the U-Pb community who use LA, ion-probe, EMPA and SEM-based techniques. The spots used by LA can completely destroy diagenetic cements, and using such techniques has to be conserved in the workflow of imaging and analysis.

"Line 605- it could result in meaningless ages- but this isn't required."

Sentence altered.

"Line 870- I would be a little careful about the wording. Pb derived from ancient source will have a high 207/206. I think you mean instead source that have a relatively high U/Pb with a long enough evolution to give lower 207/206 which is then leached or dissolved and incorporated as initial Pb in the calcite you are dating. In carbonates an easy way to do this is to somehow preserve aragonite through early diagenesis and then have it exposed to a fluid which will quickly dissolve it and take the radiogenic Pb that it produced (as well as the U)."

Yes we agree the wording has to be careful, and leaching of Pb that has sat in a ma-

terial for a long time is indeed what we mean. We have reworded this paragraph and removed the red lines from the diagram as they were confusing and potentially misleading. The aragonite example is not entirely correct – because if the U is incorporated too, then the Pb will be 'supported' and the U-Pb and Pb-Pb ratios will match and provide a young age. The process we are describing is incorporation of unsupported Pb – i.e. radiogenic lead that is decoupled from its parent U – this is how low 207Pb/206Pb and low 238U/206Pb ratios are achieved. We have added the term unsupported lead to make this clear.

"Line 906- the example of the Moab fault seems out of place. I don't understand why it follows a discussion of common Pb corrections."

This example is all about demonstrating the variability and existence of common lead compositions that are radiogenic with respect to the Stacey & Kramers model. It could be moved up, but then it would precede the discussion common lead variability to which it is pertinent.

"Line 1063 I think you mean have a high likely of not being in secular equilibrium. Your discussion at the bottom of page 42 makes a good case for ID for samples that you want to know more about secular disequilibrium in."

Line reworded (as noted by reviewer 1 also). We already state that: "This approach is based on ID, and it is unclear how effective it will be for LA-ICP-MS dating, given that 204Pb is difficult to measure at high precision. "

"Line 1289 I think you mean data instead of date."

Fixed.

"Line 1303 applied to a range of applications – should be reworded."

Fixed.

"Line 1329 record and archive say the same thing. Pick one."

Fixed.

**GChronD**

Interactive
comment

[Figure]

[Figure]

| 15 ppb | 150 ppb | 150 ppb | 500 ppb | 500 ppb | 150 ppb |

min ▮▮▮ max

**Fig. 1.** Updated Figure 1 with concentration scale

Common lead composition of vein calcite

207Pb/206Pb Intercept

Modern 'terrestrial lead'
(Stacey & Kramers, 1975)

**Fig. 2.** Updated Figure 14 without the confusing red lines

---

## Author Response (AR1)

Response to Editor's comments:

**Associate Editor Decision: Publish subject to minor revisions (further review by editor)** (02 Feb 2020) by Catherine Mottram

Comments to the Author:

Editor comments on Roberts et al., 'LA-ICP-MS U-Pb carbonate geochronology: strategies and progress with examples from fracture-fill calcite'

Overall this paper represents a useful updated review on in-situ carbonate geochronology that will be valuable contribution to the community and particularly those new to the application. It provides a thorough introduction to the method, applications and most importantly the limitations, potential pitfalls and matters of potential concern that any future analyst should be aware of.

The reviewers were largely supportive of this paper, particularly AKC and the authors have mostly addressed the specific corrections proposed by both reviewers and have presented reasonable responses to reviewers. I am mainly satisfied that these corrections have been made with the following exceptions.

There are a few points that the authors have not addressed/ might be worth considering before the manuscript is finalised:

1. Troy Rasbury commented "This manuscript is sort of a funny mix of a review and new datasets… While it is nominally a review of vein dating, it is not a complete review of what is out there." I very much agree with this opinion and I do not think that the authors have made any corrections to address this criticism. The paper is a mix of review and new data and although a lot of what is presented is very useful, it is not really a review of all previous work, instead is a compilation of all the data reported from the BGS lab. Perhaps the authors could make this clear in the abstract/introduction- or include more examples from the published literature (for instance in figures such as figure 14). Troy also commented "my overall take on the manuscript is that it could be more compactly packaged and could do a better job of reviewing all of the U/Pb dating of vein material- perhaps leaving off discussion of speleothems which is well covered by Woodhead and a laundry list of carbonate characterization techniques that aren't tied to examples." Again, I agree with this statement and I do not think that the authors have addressed this- they have made no attempt to make the manuscript any shorter or more compact. I have two suggestions for how you might achieve this:

We appreciate the comments from those not involved the manuscript, since they can provide an unbiased opinion of the paper. We also understand the general feeling gained by the reviewers and editor – that this paper presents a mixture of review and new data. We however, do not feel that a paper should conform to a simple 'research article' or a 'review paper', these distinctions are given to us by the publishers, and should not limit how we want to present science and scientific methodology. But, we appreciate that the comments may improve the paper, and thus, we have attempted to improve the paper by shortening some aspects, merging some sections, and removing some duplication of text.

Although most of our examples draw from vein material, all of our points of discussion are relevant to all carbonate types, and thus we have reworded any text to make it clear that this paper applies to all applications.

Specifically, we have completely redrawn the figure with the imaging techniques, which we feel provides a more useful tableau, and which is better linked to the text. We have merged the three 'examples' of image-guided dating into a single figure, and dramatically reduced the text, removing any unnecessary background information about the samples. We have removed one of the three image-based dating examples. We have removed the figure with the mu values compiled (but added it to the supp file). We have removed the discussion section and expanded the conclusions, and added all the limitations into a single 'limitation' section. This still comes towards the end of the paper, but we stand by our opinion that is only natural to discuss 'the ways doing things', some 'examples of doing things', and then some 'nuances when doing those things'.

Why not review the published literature as a whole? – (1) if we were to swap figure 4 (the isochron good, bad and ugly) for published literature only, then this section would essentially be unpicking other peoples work (not ideal); (2) It would be difficult to draw from the literature for the section on image-guided dating, as there is very little information within the published literature on how some dates may have been refined, selected, interpreted etc. Instead, we have chosen to include unpublished data, such that we can better demonstrate the methods behind refining data with imagery/compositional data; (3) We have included image-based dating examples from veins, as these are complementary to those published by Drost et al on sedimentary carbonates, and are a better comparison with the image-guided examples also on veins.
Hopefully it should be clear where all the data are drawn from, i.e. literature or our own data.

a. Section 2- Discussion of ID vs in-situ techniques- although this is of course a very important topic for discussion in geochronology much of this discussion has already been covered in other publications focused on geochronology methods. I therefore think this text could be shortened. Figure 2 for instance is very useful and I think if the text could be focused on these carbonate-specific examples would help concentrate the discussion.
We find that many papers cover the basics of ID versus LA, i.e. the difference in spatial resolution and in precision and accuracy of data, but they do not cover the nuances that make LA so beneficial for carbonates, for example, the ability to hit zones with different U/Pb ratios. We have removed a few sentences that were unnecessary, but overall we have maintained this important section.

b. Section 8- Troy's comment "The modeling of initial disequilibrium is a very important contribution. Why is it so late in the paper?" I agree with this- I think that this section is an important and relatively novel contribution to the community, however, it is buried at the end of this very long paper and I worry that it will be lost due to this. I think that this would make much more impact if this was taken out and made into its own separate manuscript. You could then extend to include details of how to make the corrections with worked examples. This would help the reader understand how to make the corrections as well as why they
are important. It is up to the authors what to do with this section- I realise that you won't want to make it
into a separate manuscript, but I just wanted you to know that I think it would be much more widely read
if you did separate it. Perhaps you could consider adding a worked example as a small addition or refer to
an example from the literature?
We have kept this section in this paper. Why? We feel that we should not be drawn into the routine of
splitting papers up into smaller parts, it can be good for metrics, but generally doesn't make great papers.
This particular discussion of diseq is rather immature, certainly not the full story and any answers. It is
clearly important to point out, but how we tackle the uncertainty of activity ratios, and how we incorporate
these uncertainties into age uncertainties, is not clear. This in fact requires a community-led approach, for
example as the zircon and U-Th working groups have worked in the past. Thus, we feel this section is
merely paving the way for new investigation and appreciation of a problem. Could it be moved forward
within the paper?, yes it could come before other sections, but we do not feel 13500 words is a particularly
lengthy paper such that readers will not make it to the limitations section.
In addition, Noah McLean, a co-author, is writing up the equations behind integration of U-Pb and U-Th
data to incorporate activity ratio measurements – which hopefully will make it into the same special issue.
This paper will be more technical, and as such, we feel the broad discussion in the present paper is best
staying where it is.
2. Although the methods are clearly presented in the supplementary material, it is not clear whether the
primary data for any unpublished examples used by the authors is presented in the Supplementary
material? Perhaps the authors could include a data table for any unpublished data?
All unpublished data behind the presented isochrons in the image-guided and image-based sections are
now included in the supplementary files.
I have also made a few suggestions on the attached annotated manuscript.
We have made some of these suggestion in our shortening.

[revised manuscript text omitted]